# A counter gradient of Activin A and follistatin instructs the timing of hair cell differentiation in the murine cochlea

Meenakshi Prajapati-DiNubila[1,2†], Ana Benito-Gonzalez[1,2†], Erin Jennifer Golden[1,2‡], Shuran Zhang[1,2], Angelika Doetzlhofer[1,2]*

[1]Solomon H. Snyder Department of Neuroscience, Johns Hopkins University School of Medicine, Baltimore, United States; [2]Center for Sensory Biology, Johns Hopkins University School of Medicine, Baltimore, United States

*For correspondence:
adoetzlhofer@jhmi.edu

[†]These authors contributed equally to this work

Present address: [‡]Department of Cell and Developmental Biology, The Rocky Mountain Taste and Smell Center University of Colorado, Aurora, United States

Competing interests: The authors declare that no competing interests exist.

**Abstract** The mammalian auditory sensory epithelium has one of the most stereotyped cellular patterns known in vertebrates. Mechano-sensory hair cells are arranged in precise rows, with one row of inner and three rows of outer hair cells spanning the length of the spiral-shaped sensory epithelium. Aiding such precise cellular patterning, differentiation of the auditory sensory epithelium is precisely timed and follows a steep longitudinal gradient. The molecular signals that promote auditory sensory differentiation and instruct its graded pattern are largely unknown. Here, we identify Activin A and its antagonist follistatin as key regulators of hair cell differentiation and show, using mouse genetic approaches, that a local gradient of Activin A signaling within the auditory sensory epithelium times the longitudinal gradient of hair cell differentiation. Furthermore, we provide evidence that Activin-type signaling regulates a radial gradient of terminal mitosis within the auditory sensory epithelium, which constitutes a novel mechanism for limiting the number of inner hair cells being produced.

DOI: https://doi.org/10.7554/eLife.47613.001

## Introduction

Housed in the inner ear cochlea, the auditory sensory organ contains a spiral shaped sensory epithelium specialized to detect and transduce sound. Along its longitudinal axis, two types of mechano-sensory cells, referred to as inner and outer hair cells, are arranged in distinct rows, with three rows of outer hair cells and a single row of inner hair cells. To ensure the highly stereotyped arrangement of hair cells, cell cycle withdrawal and differentiation within the auditory sensory epithelium occurs in a spatially and temporally highly coordinated manner. Auditory sensory progenitors (pro-sensory cells) exit the cell cycle in an apical-to-basal gradient (*Chen et al., 2002*; *Lee et al., 2006*; *Ruben, 1967*), whereas their differentiation into hair cells and supporting cells occurs in an opposing, basal-to-apical gradient (*Chen et al., 2002*). Recent studies uncovered that Sonic Hedgehog (SHH) signaling plays a key role in setting up the spatial and temporal pattern of auditory hair cell differentiation. In the undifferentiated cochlea, pro-sensory cells are exposed to high concentrations of SHH protein secreted by the adjacent spiral ganglion neurons (SGNs) (*Bok et al., 2013*; *Liu et al., 2010*). SGN-specific ablation of SHH or cochlear epithelial-specific ablation of its co-receptor Smoothened results in premature hair cell differentiation and in the most extreme case a reversal of the gradient of hair cell differentiation (*Bok et al., 2013*; *Tateya et al., 2013*). The basic-helix-loop-helix (bHLH) transcription factor ATOH1 is the earliest marker of nascent hair cells and is both necessary and sufficient for the production of hair cells (*Bermingham et al., 1999*; *Cai et al., 2013*; *Woods et al., 2004*; *Zheng and Gao, 2000*). SHH signaling represses hair cell differentiation, at least in part, through maintaining the expression of HEY1 and HEY2 in pro-sensory cells (*Benito-*

*Gonzalez and Doetzlhofer, 2014*; *Tateya et al., 2013*). The bHLH transcriptional repressors HEY1 and HEY2 have been shown to antagonize ATOH1 function (*Doetzlhofer et al., 2009*) and their loss leads to premature hair cell differentiation (*Benito-Gonzalez and Doetzlhofer, 2014*). In addition to HEY1 and HEY2, auditory pro-sensory cells express members of the inhibitor of differentiation (ID) family (*Jones et al., 2006*). ID proteins are dominant negative regulators of bHLH transcription factors (reviewed in *Wang and Baker, 2015*). Acting downstream of BMP signaling, ID1-3 are thought to maintain pro-sensory cells in an undifferentiated state by interfering with ATOH1's ability to bind to DNA and auto-regulate its own expression (*Jones et al., 2006*; *Kamaid et al., 2010*). Through a yet unknown mechanism, *Id1-3* expression is downregulated in a subset of pro-sensory cells at the onset of differentiation, allowing these cells to upregulate ATOH1 and to differentiate into hair cells.

Much less is known about the signals and factors that promote ATOH1 expression/activity within pro-sensory cells and their role in auditory hair cell differentiation. Over-activation of Wnt/β-catenin signaling has been shown to increase *Atoh1* expression in differentiating cochlear explants, and in the absence of Wnt/β-catenin signaling hair cells fail to form (*Jacques et al., 2012*; *Munnamalai and Fekete, 2016*) (*Shi et al., 2014*). However, the pattern of Wnt–reporter activity, which at the onset of hair cell differentiation is high in the cochlear apex but low in the cochlear base, does not parallel the basal-to-apical wave of differentiation (*Jacques et al., 2012*). Interestingly, the *Inhba* gene, which encodes the Activin A subunit Inhibin βA (*Barton et al., 1989*), has been recently reported to be expressed in a basal-to-apical gradient within the differentiating auditory sensory epithelium (*Son et al., 2015*). Activins, which belong to the transforming growth factor (TGF)-β superfamily of cytokines, control a broad range of biological processes, including reproduction, embryonic axial specification, organogenesis and adult tissue homeostasis (reviewed in *Namwanje and Brown, 2016*). Canonical TGFβ-type signaling is transduced by receptor regulated SMAD proteins (R-SMADs). Upon receptor mediated phosphorylation, R-SMADs (SMAD1, 2, 3, 5, 9) form heteromeric complexes with SMAD4, which enables them to translocate to the nucleus and activate a broad array of target genes (reviewed in *Massagué, 2012*). In the developing spinal cord, Activins and other TGF-β-related ligands are required in most dorsally located neuronal progenitors for *Atoh1* induction and their subsequent differentiation as D1A/B commissural neurons (*Lee et al., 1998*; *Wine-Lee et al., 2004*). The role of Activin-type signaling in cochlear *Atoh1* regulation and hair cell differentiation is currently unknown.

Here, we identify Activin A and its antagonist follistatin (FST) as key regulators of *Atoh1* gene expression and hair cell differentiation. We find that in the developing murine cochlea Activin A acts as a pro-differentiation signal, and demonstrate that a counter gradient of Activin A and FST within the auditory sensory epithelium times the basal-to-apical wave of hair cell differentiation. Furthermore, we provide evidence that a counter gradient of Activin A and FST informs a previously unidentified medial-to-lateral gradient of terminal mitosis that forces inner hair cell progenitors located at the medial edge of the sensory epithelium to withdraw from the cell cycle prior to outer hair cell progenitors.

## Results

### The graded pattern of Activin A expression parallels auditory hair cell differentiation

The biological activity of Activins and other Activin-type ligands is limited by the secreted protein follistatin (FST). Two FST molecules encircle the Inhibin β dimer, blocking both type I and type II receptor binding sites, thus preventing receptor binding and activation of its downstream signaling cascade (*Thompson et al., 2005*). Within the differentiating auditory sensory epithelium *Fst* and the Inhibin βA encoding gene *Inhba* are expressed in opposing gradients, with *Inhba* being highest expressed within the basal sensory epithelium and *Fst* being highest expressed apically (*Son et al., 2015*). To explore a potential correlation with hair cell differentiation we analyzed the pattern of *Atoh1*, *Inhba* and *Fst* mRNA expression in the developing cochlea stages E13.5-E15.5 (*Figure 1A–C*). In mice, *Atoh1* expression starts around embryonic stage E13.5-E14.0 in a subset of *Sox2* positive pro-sensory cells at the basal turn of the cochlea (*Figure 1A*). Paralleling *Atoh1* expression, *Inhba* expression was limited to the basal pro-sensory domain. In contrast, *Fst* was highly expressed within the lateral part of the pro-sensory domain throughout the cochlear apex and mid turn but was only

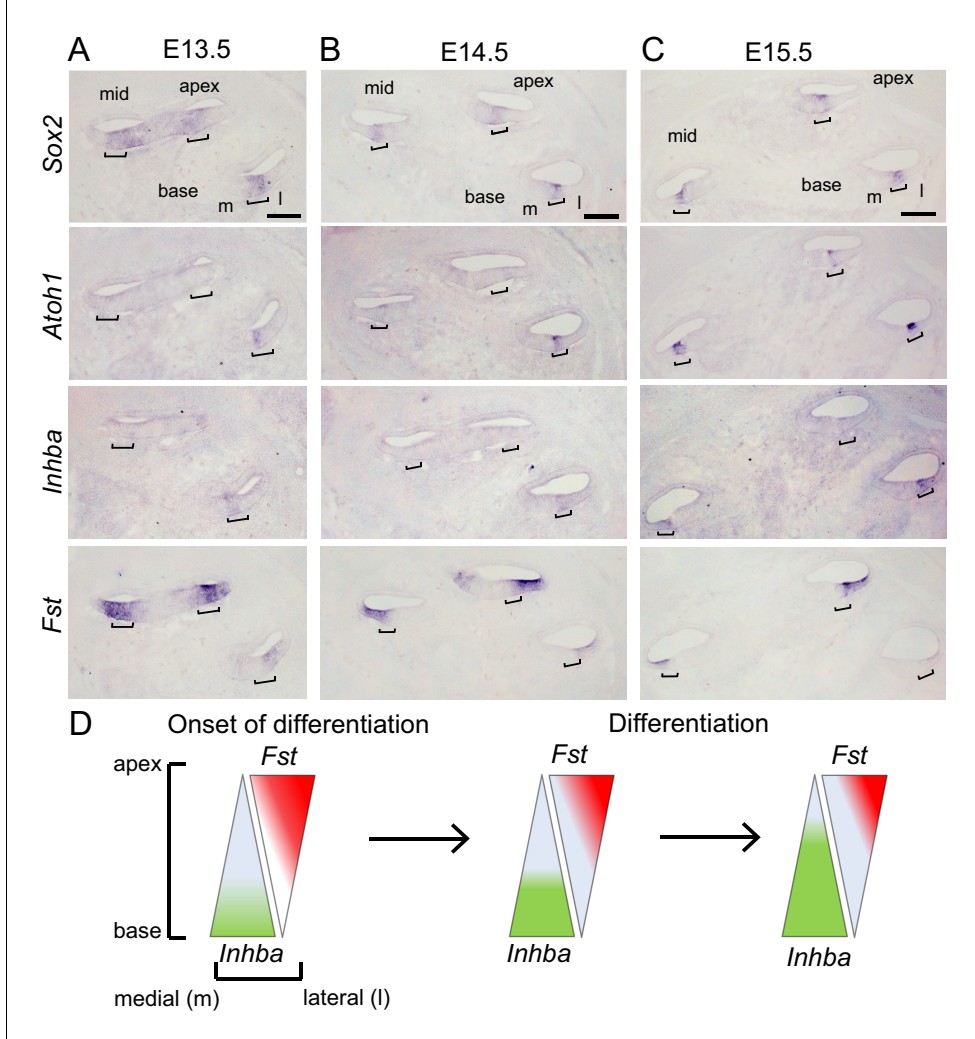

**Figure 1.** Activin A expression parallels auditory hair cell differentiation. (A-C) In situ hybridization (ISH) was used to analyze the cochlear expression pattern of *Inhba* and *Fst* at the onset of (A, E13.5) and during hair cell differentiation (B, E14.5 and C, E15.5) in adjacent serial cochlear sections. *Sox2* transcript marks the pro-sensory/sensory domains, *Atoh1* transcript marks nascent hair cells. Brackets mark the pro-sensory/sensory domains with the cochlear duct. Abbreviations: m, medial; l, lateral. Scale bar, 100 µm. (D) Schematics of longitudinal (apical–basal) and radial (medial-lateral) expression gradients of *Inhba* (green) and *Fst* (red) within the auditory sensory epithelium.

DOI: https://doi.org/10.7554/eLife.47613.002

weakly expressed in the cochlear base. At stages E14.5 and E15.5, as hair cell differentiation progressed towards the cochlear apex, *Inhba* expression within the pro-sensory/sensory domain extended to the cochlear mid-turn (*Figure 1B,C*). At the same time, *Fst* expression further regressed in the cochlear base and weakened in the cochlear mid-turn but continued to be highly expressed in the undifferentiated cochlear apex.

As summarized in *Figure 1D*, our analysis of *Inhba* and *Fst* expression suggests that pro-sensory cells in the differentiating cochlea are exposed to an Activin A signaling gradient that closely resembles the spatial and temporal pattern of hair cell differentiation.

## Activin A promotes precocious cochlear hair cell differentiation

Activins play a central role in coordinating growth and differentiation in sensory and neuronal tissues, including the retina, olfactory epithelium and spinal cord (*Davis et al., 2000*; *Gokoffski et al.,*

*2011*; *Liem et al., 1997*). To determine the role of Activin A in auditory hair cell differentiation, we exposed undifferentiated cochlear tissue (stage E13.5) to recombinant Activin A for 24 hours (hr) (*Figure 2A*). As a readout for hair cell differentiation we used transgenic cochlear tissue that expressed nuclear localized green fluorescent protein (GFP) under the control of the *Atoh1* enhancer (*Lumpkin et al., 2003*). Hair cell differentiation follows a steep basal-to-apical gradient in which hair cells located within the cochlear mid-basal segment differentiate prior to more apically located hair cells. In addition, a shallower, medial-to-lateral gradient exists, with inner hair cells differentiating prior to their neighboring more laterally located outer hair cells (*Chen et al., 2002*). After 12 hr of culture, control cultures showed no or only weak *Atoh1-GFP*-reporter activity (GFP) within the developing cochlear duct (*Figure 2B*, *Control*, 12 hr). By contrast, a narrow stripe of GFP positive inner hair cells was already evident in Activin A-treated cochlear explants (*Figure 2B*, *Activin A*, 12 hr). 6 hr later, a broad stripe of GFP positive cells representing inner and outer hair cells was present in Activin A-treated cochlear explants, whereas control cochlear explants contained only few scattered GFP positive inner hair cells (*Figure 2B*, 18 hr). Furthermore, at all stages examined, the band of GFP positive cells extended further apically in Activin A-treated cochlear explants compared to control cochlear explants (*Figure 2B,C*). Taken together, our findings indicate that Activin A acts a differentiation signal for auditory hair cells.

To identify downstream targets of Activin signaling in the developing cochlea, we cultured E13.5 cochlear explants with or without Activin A for 24 hr, after which we enzymatically purified the corresponding cochlear epithelia and performed RT-qPCR (*Figure 2D*). Our analysis focused on known effectors of hair cell differentiation and their upstream regulators. We found that Activin A treatment increased the expression of *Atoh1* and the Wnt-target gene *Lgr5* by more than 4-fold and increased the expression of *Pou4f3*, a critical early hair cell-specific transcription factor by more than 12-fold. Furthermore, Activin treatment led to a more than 2-fold reduction in *Bmp4, Id2, Id3 and Id4* expression, but did not significantly alter the expression of other pro-sensory genes (p27/Kip1- official gene name *Cdkn1b*, *Etv5*, *Hey1*, *Hey2*, *Fgf20* and *Ptch1*).

BMP receptor signaling has been shown to positively regulate *Id1-3* expression in cochlear and vestibular pro-sensory cells (*Kamaid et al., 2010*; *Ohyama et al., 2010*). However, the role of BMP signaling in cochlear *Id4* gene expression has not yet been established. To further characterize the role of BMP and Activin signaling in *Atoh1* and *Id* gene regulation, we cultured enzymatically purified cochlear epithelia for 3.5 hr in the presence of either Activin A, BMP4 or vehicle control (*Figure 2E*). Brief BMP4 treatment significantly increased *Id1 and Id3* expression compared to control but had no effect on *Id4 and Atoh1* expression (*Figure 2F*). Conversely, the brief Activin A treatment resulted in a modest, but significant decrease in *Id4* expression compared to control but did not alter *Id1-3* or *Atoh1* expression (*Figure 2F*). The divergent transcriptional responses to Activin A and BMP4 treatment are likely the consequence of differences in R-SMAD utilization. Activation of Activin signaling commonly leads to the phosphorylation and activation of SMAD2 and SMAD3, whereas stimulation of BMP signaling leads to the phosphorylation and activation of SMAD1, SMAD5 and SMAD9 (reviewed in *Miyazawa et al., 2002*). Indeed, we found that in the developing cochlea Activin A selectively stimulated the phosphorylation of SMAD2/3, whereas BMP4 treatment selectively induced the phosphorylation of SMAD1/5/9 (*Figure 2G*).

Taken together these findings suggests that Activin A positively regulates *Atoh1* expression through an indirect mechanism that, at least in part, involves the repression of BMP-dependent and independent *Id* gene expression.

## FST overexpression delays cochlear hair cell differentiation

To characterize the function of FST in cochlear development we made use of a previously developed doxycycline (dox) inducible transgenic mouse line, in which a cassette encoding the human FST-288 isoform is under the control of a tetracycline-responsive promoter element (tetO) (*Lee and McPherron, 2001*; *Roby et al., 2012*) (*Figure 3A*). In the presence of a ubiquitously expressed reverse tetracycline-controlled trans-activator (rtTA), dox administration allows for robust induction of the human (h) FST transgene (*Figure 3A,B*). To assess the role of FST in cochlear hair cell differentiation, timed pregnant females received dox beginning at E11.5 and double transgenic FST over-expressing embryos (*R26-FST*) and single transgenic littermates lacking the R26-M2rtTA transgene (control) were harvested three days later (E14.5). Inclusion of the *Atoh1*-reporter transgene (*Atoh1-GFP*) allowed for the ready analysis of hair cell differentiation at the time of isolation. While control

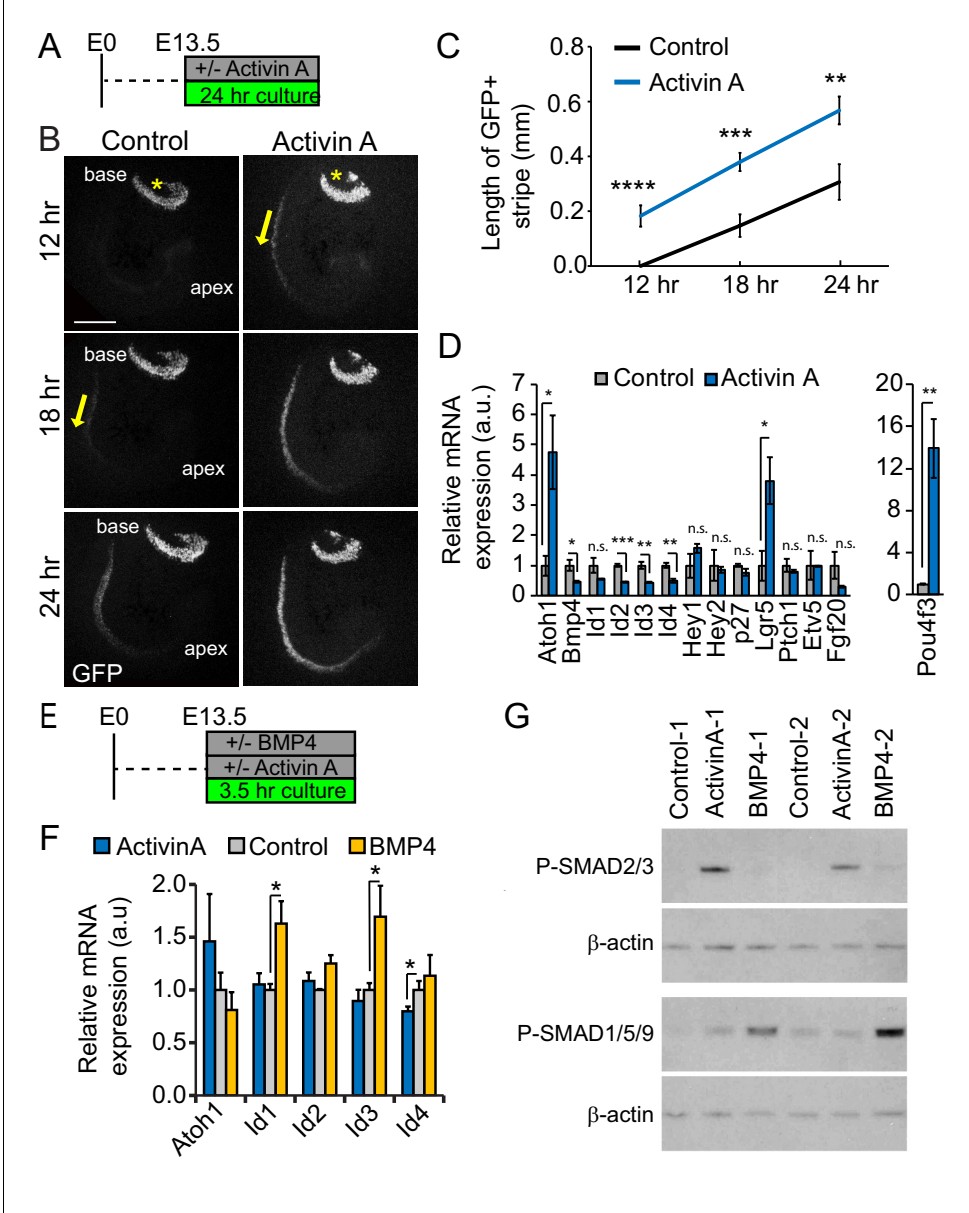

**Figure 2.** Activin A promotes auditory hair cell differentiation. (**A**) Experimental design for B-D. Stage E13.5 wild type cochlear explants were cultured with or without Activin A (final conc. 500 ng/ml) for 24 hr. (**B**) *Atoh1-GFP* reporter expression (GFP, gray) was used to monitor and analyze hair cell differentiation in Activin A-treated and control cochlear explants. Asterisks mark the vestibular saccule that contain GFP positive hair cells. Yellow arrows mark the onset of hair cell differentiation within the cochlea. Scale bar, 100 µm. (**C**) Quantification of basal-to-apical extent of hair cell differentiation in control versus Activin A-treated cochlear cultures (**B**). Data expressed as mean ± SEM (n = 5–8 cochlear explants per group, **p<0.01, ***p<0.001, ****p<0.0001, student's t-test). (**D**) Transcript levels of pro-sensory genes (*Id1-4, Hey1, Hey2*, p27, *Etv5, Fgf20, Ptch1*) and hair cell-specific genes (*Atoh1, Pou4f3*) were analyzed in enzymatically purified cochlear epithelia. Data are mean ± SEM (n = 3 biological replicates, *p<0.05, **p<0.01, ***p<0.001, student's t-test). (**E**) Experimental design for F, G. Stage E13.5 wild type cochlear epithelia were cultured with or without Activin A (final conc. 200 ng/ml) or BMP4 (final conc. 100 ng/ml) for 3.5 hr. (**F**) RT-qPCR analysis reveals differential response to Activin A and BMP4 treatment. Individual cochlear epithelia were analyzed. Data are mean ± SEM, n = 4 biological replicates, *p<0.05. (**G**) Activin A induces SMAD2/3 phosphorylation in cochlear epithelial cells. Western blot analysis was used to detect phosphorylated (p) SMAD2/3 and p-SMAD1/5/9 proteins in individual cochlear epithelia after 3.5 hr culture with or without Activin A or BMP4. Beta-actin was used as loading control.

DOI: https://doi.org/10.7554/eLife.47613.003

cochlear tissue already contained inner hair cells at the base of the cochlea, as shown here using cochlear whole mounts and sections, FST overexpressing cochlear tissue still lacked hair cells (*Figure 3C,D*).

Consistent with FST antagonizing hair cell differentiation, we found that *Atoh1* and *Pou4f3* mRNA expression was significantly reduced in E14.5 FST overexpressing cochlear epithelia, whereas the mRNA expression of *Id3* and *Id4* was significantly increased compared to control (*Figure 3E*). In addition, FST overexpression resulted in a modest but significant increase in the expression of the SHH receptor encoding gene *Ptch1* (*Driver et al., 2008*) and the FGFR target and effector gene *Etv5* (*Hayashi et al., 2008*). Despite these changes, the expression of *Fgf20, Hey1* and *Hey2*, which are targets and effectors of SHH/FGFR signaling (*Benito-Gonzalez and Doetzlhofer, 2014*; *Tateya et al., 2013*), remained unchanged in response to FST overexpression (*Figure 3E*). Interestingly, FST overexpression also led to a modest but significant reduction in the expression of *Bmp4*. To investigate whether FST overexpression selectively disrupted Activin and/or BMP signaling, we analyzed P-SMAD2/3 and P-SMAD1/5/9 protein levels in cochlear epithelial protein extracts obtained from E14.5 FST overexpressing embryos and their control littermates. Consistent with its

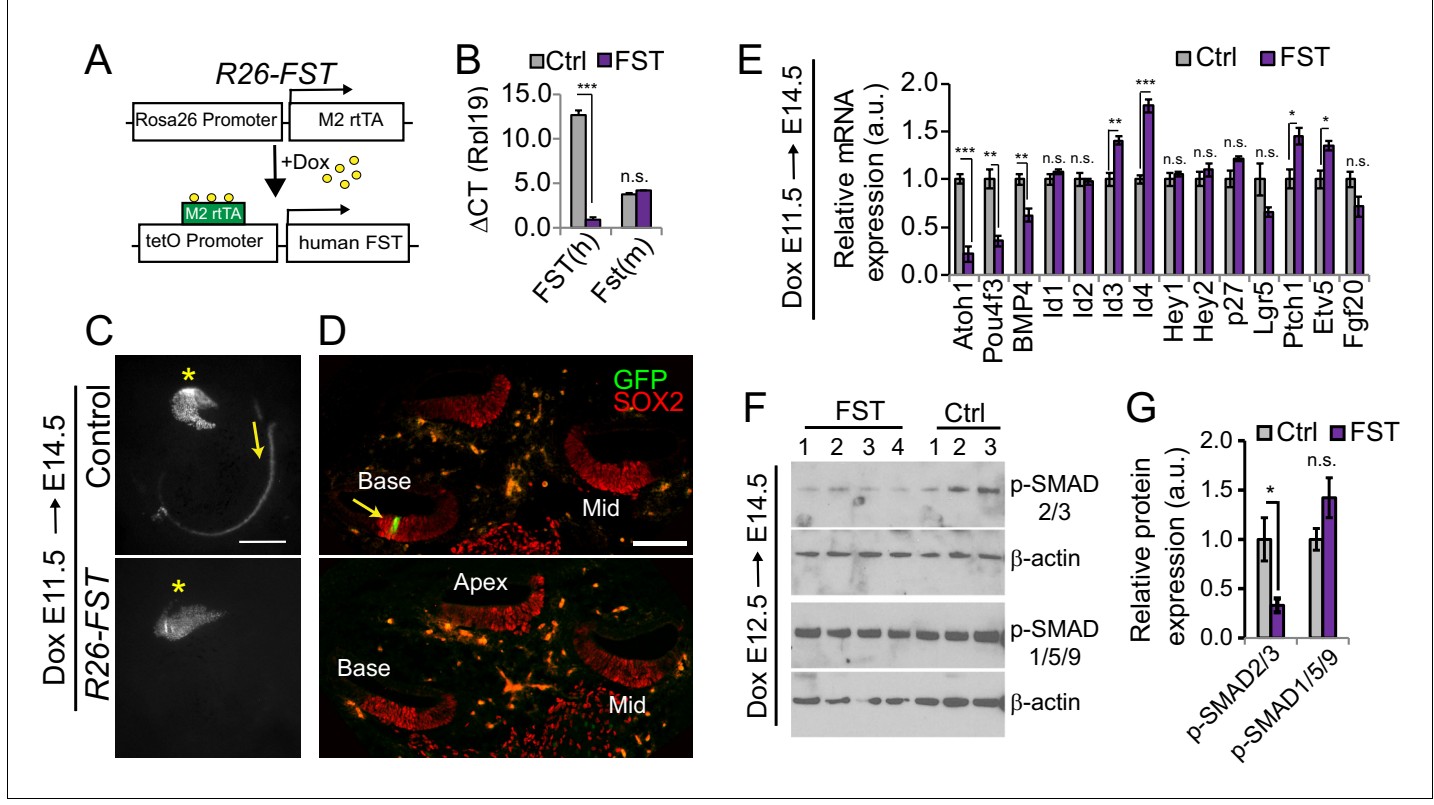

**Figure 3.** FST overexpression interferes with auditory hair cell differentiation. (**A**) Inducible *FST* transgenic mouse model. In the presence of doxycycline (dox) double transgenic animals (*R26-M2rtTA* and *tetO-human FST*) express human FST under the control of the R26 promoter (*R26-FST*). Non-transgenic littermates and littermates that carry only one of the transgenes were used as experimental controls (Ctrl). (**B**) Human (h) *FST* transgene expression in control (Ctrl) and *R26-FST* transgenic (FST) cochlear epithelia after 48 hr of dox administration. Plotted is the difference in cycle threshold (ΔCT) compared to the reference gene *Rpl19*. Data expressed as mean ± SEM (n = 4 animals per group, ***p≤0.001, student's t-test). (**C**) Low power fluorescent images showing native *Atoh1-GFP* reporter expression (GFP, gray) in wild type (control) and FST overexpressing (*R26-FST*) cochleae stage E14.5. Asterisks mark the vestibular saccule that contain GFP positive hair cells. Yellow arrows mark the onset of hair cell differentiation within the cochlea. Scale bar 100 μm. (**D**) Confocal images of wild type (control) and FST overexpressing (*R26-FST*) cochlear cross sections, stage E14.5. GFP expression (GFP, green) marks hair cells (yellow arrow), SOX2 staining (red) marks the sensory domain. Scale bar 100 μm. (**E**) RT-qPCR-based analysis of gene expression in FST overexpressing (FST) and control cochlear epithelia (Ctrl). Data are mean ± SEM (n = 4–5 animals per group, *p<0.05, **p<0.01, ***p<0.001, student's t-test). (**F**) Western blot-based analysis of p-SMAD2/3 and p-SMAD1/5/9 protein expression in stage E14.5 *FST* transgenic (FST: 1–4) and control (Ctrl: 1–3) cochlear epithelia. Beta-actin was used as loading control. (**G**) Quantification of p-SMAD2/3 and p-SMAD1/5/9 protein levels in F.

DOI: https://doi.org/10.7554/eLife.47613.004

role as a canonical Activin antagonist, overexpression of FST significantly reduced P-SMAD2/3 protein levels in the developing cochlea. By contrast, cochlear P-SMAD1/5/9 protein levels, a readout for BMP signaling, remained unchanged in response to FST overexpression (*Figure 3F,G*), suggesting that FST overexpression does not disrupt BMP signaling in the developing cochlea.

To address whether Activin A can rescue the FST mediated delay in hair cell differentiation, we harvested cochlear tissue from stage E13.5 FST overexpressing (*R26-FST*) embryos and their single transgenic wild type littermates, cultured them with or without Activin A and monitored hair cell differentiation for the next 48 hr (*Figure 4A*). Consistent with our earlier findings, we observed that hair cell differentiation occurred significantly earlier in Activin A-treated wild type cochlear explants compared to untreated wild type cochlear explants (*Figure 4B, C,F*). Conversely, the onset of hair cell differentiation was significantly delayed in untreated FST overexpressing cochlear explants compared to untreated wild type cochlear explants (*Figure 4B, D,F*). These defects were almost completely abolished when Activin A treatment and FST overexpression were combined and no significant differences in the onset or in the progression of hair cell differentiation was observed between FST overexpressing cochlear explants treated with Activin A and wild type untreated cochlear explants (*Figure 4B,E,F*). In summary, our data suggest that FST antagonizes hair cell differentiation in an Activin A-dependent manner.

## Disruption of Activin signaling results in ectopic inner hair cell formation

Next, we analyzed whether FST overexpression altered the cellular patterning of the sensory epithelium. Induction of FST overexpression at E11.5 or as late as E14.5 resulted in neonatal lethality of *R26-FST* transgenic animals, limiting our analysis to late embryonic stages. We used native GFP (*Atoh1-GFP*) expression, myosin VII a (Myo7a) immunostaining and phalloidin labeling to identify hair cells and SOX2 immunostaining to identify supporting cells. As expected, cochlear tissue from stage E18.5 control embryos contained a single row of inner hair cells and three rows of outer hair cells (*Figure 5A,B*, *Control*). By contrast, *R26-FST* cochlear tissue contained two rows of inner hair cells throughout the basal half of the cochlea (*Figure 5A,B*, *R26-FST*). The observed ectopic inner hair cells were unlikely to be the product of supporting cell-to-hair cell fate conversion, as the density of supporting cells that surrounded inner hair cells-called inner phalangeal cells- was similarly increased in response to

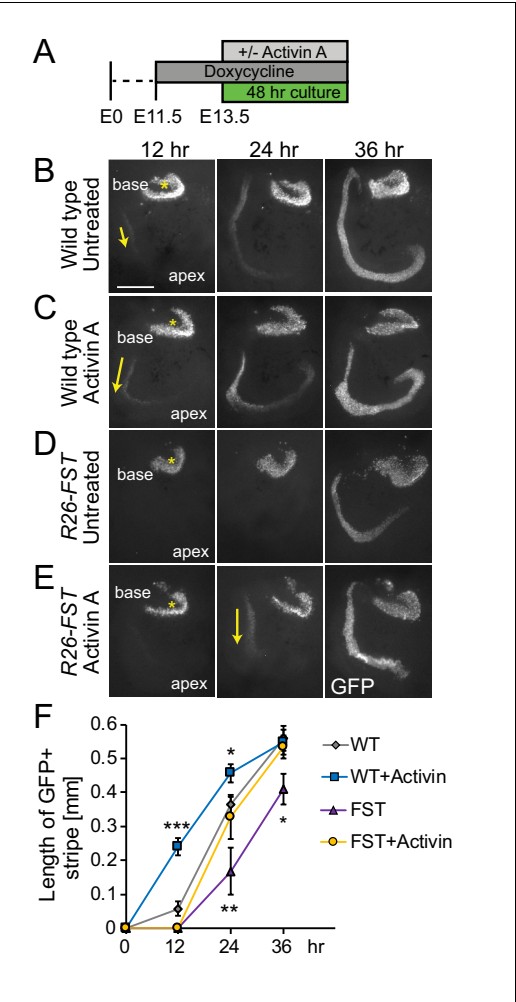

**Figure 4.** Exogenous Activin A rescues the FST induced delay in auditory hair cell differentiation. (**A**) Experimental design for B-E. Dox was administered to timed pregnant dams starting at E11.5. At E13.5, cochlear tissue from FST overexpressing embryos (*R26-FST*) and wild type littermates were cultured for 48 hr with or without Activin A (500 ng/ml). (**B–E**) *Atoh1-GFP* reporter expression (GFP, gray) marks nascent hair cells. Asterisks indicate hair cells within vestibular sacculus. Yellow arrows mark nascent cochlear hair cells. Scale bar, 100 μm. (**F**) The length of the GFP positive sensory epithelium was used to quantify the extent of hair cell differentiation in wild type (WT) and FST overexpressing (FST) cochlear explants cultured with and without Activin A. Data expressed as mean ± SEM (n = 5–8 cochlear explants per group, *p≤0.05, **p<0.01, ***p<0.001, student's t-test). Two independent experiments were conducted and data compiled.

DOI: https://doi.org/10.7554/eLife.47613.005

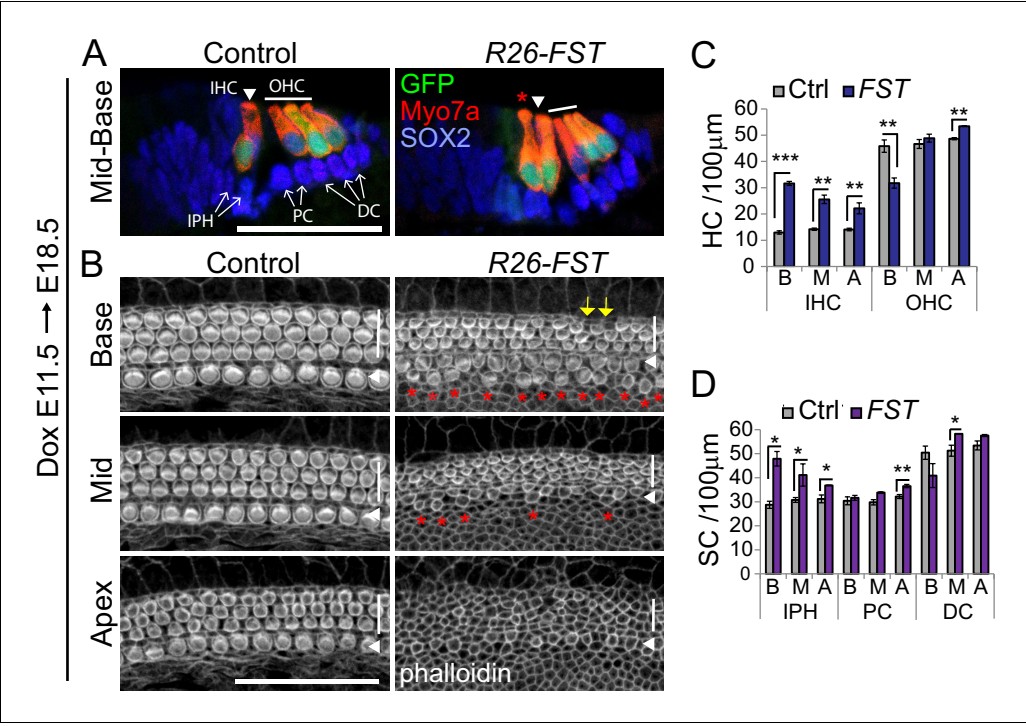

**Figure 5.** FST overexpression in the developing cochlea disrupts inner hair cell patterning and delays hair cell maturation. *FST* transgenic (*R26-FST*) embryos and their control (wild type or single transgenic) littermates were exposed to dox starting at E11.5 until tissue harvest at E18.5. (**A**) FST overexpression results in ectopic inner hair cells. Shown are cross-sections through the cochlear mid-base of control and *R26-FST* transgenic embryos. GFP (green) and Myo7a (red) label inner hair cells (IHC, white arrowhead) and outer hair cells (OHC, white bar). Red asterisks mark ectopic inner hair cells. SOX2 (blue) labels supporting cells including inner phalangeal cells (IPH), pillar cells (PC) and Deiters' cells (DC) indicated by white arrows. Scale bar 50 µm. (**B**) FST overexpression delays stereocilia formation. Shown are z-stack projections of the luminal surface of control and *R26-FST* transgenic cochlear sensory epithelia. Phalloidin labels actin-rich stereocilia of inner (white arrowhead) and outer hair cells (white bar). Red asterisks mark ectopic inner hair cells. Yellow arrows mark the location of missing outer hair cells. Scale bar 50 µm. (**C–D**) Quantification of hair cell (**C**) and supporting cell (**D**) density in the base, mid and apex of control (Ctrl, gray bars) and FST overexpressing (FST, purple bars) cochleae. Abbreviations: IHC, inner hair cells; OHC, outer hair cells; IPH, inner phalangeal cells; PC, pillar cells; DC, Deiters'; B, base; M, mid; A, apex. Data expressed as mean ± SEM (n = 3 animals per group, *p≤0.05, **p<0.01, student's t-test).
DOI: https://doi.org/10.7554/eLife.47613.006

The following figure supplement is available for figure 5:

**Figure supplement 1.** FST overexpression delays the thinning of the auditory sensory epithelium.
DOI: https://doi.org/10.7554/eLife.47613.007

FST overexpression (*Figure 5C,D*). Interestingly, the outer hair cell density and cellular patterning was unchanged in response to FST overexpression except for the cochlear base. The base of FST overexpressing cochleae contained short stretches of sensory epithelium in which the third row of outer hair cells was missing, resulting in a modest reduction in outer hair cell density (*Figure 5B,C*). Furthermore, differentiation/maturation of the auditory sensory epithelium was severely delayed in response to FST overexpression. Usually, at stage E18.5, the basal-to-apical wave of differentiation has reached the apex and the initially multilayered auditory sensory epithelium has thinned to a two-layered sensory epithelium, and is near its final length (*Driver et al., 2017*). Moreover, hair cells throughout the cochlea have formed actin-rich apical protrusions, referred to as stereocilia (*Son et al., 2012*) (*Figure 5B*, Control). Consistent with a severe delay in hair cell differentiation, auditory sensory epithelia from *R26-FST* transgenic animals had not yet thinned to a single layer of supporting cells (*Figure 5—figure supplement 1*) and their length was 30% shorter compared to auditory sensory epithelia from control littermates (single transgenic control = 5.14 ± 0.04 mm, *R26-*

*FST* = 3.65 ± 0.13 mm, n = 5, p=0.0002). Moreover, hair cells in FST overexpressing cochlear tissue had less mature stereocilia than their wild type counterparts, and apical hair cells lacked stereocilia completely (*Figure 5B*, *R26-FST*).

FST induction one day later at ~E13.0 (dox E12.5), had no effect on the length of the auditory sensory epithelium (dox E12.5, harvest E17.5: single transgenic control = 4.13 ± 0.17 mm, *R26-FST* = 3.99 ± 0.12 mm, n = 3, p=0.5425). However, we continued to observe ectopic inner hair cells in FST overexpressing cochleae, with the highest number of ectopic hair cells found in the basal portion of the auditory sensory epithelium, as well as short stretches of sensory epithelium with only two rows of outer hair cells (*Figure 6A,B*), indicating that the FST-induced defects in hair cell patterning were not the result of stunted cochlear outgrowth (extension). Furthermore, we found that hair cell stereocilia in FST overexpressing cochleae had a less mature phenotype compared to their counterparts in control cochlear tissue, indicating that FST induction as late as ~E13.0 causes a delay in hair cells differentiation/maturation (*Figure 6A*).

To determine whether FST influences hair cell differentiation and patterning in an Activin A-dependent manner, we analyzed hair cell morphology and patterning in Activin A (*Inhba*) mutant mice. To selectively ablate *Inhba* gene function in the developing inner ear, we intercrossed *Inhba* floxed mice , in which exon 2 of the *Inhba* gene is flanked by LoxP sites (*Pangas et al., 2007*), with inner ear-specific *Pax2-Cre* transgenic mice (*Pax2-Cre; Inhba fl/fl*) (*Ohyama and Groves, 2004*). The conditional knockout (cKO) of the *Inhba* gene did not alter overall cochlear morphology, nor did it alter the length of the sensory epithelium compared to control (*Inhba fl/fl*) (P0: control = 4.99 ± 0.12 mm; *Inhba* cKO = 4. 82 ± 0.01 mm; n = 2, p-value=0.4043).

However, as predicted, loss of *Inhba* gene function resulted in hair cell maturation and patterning defects. The observed defects were much milder, but qualitatively similar to the defects observed in response to FST overexpression (dox E12.5). Cochlear tissue from control littermates (*Inhba fl/fl*) contained the normal complement of one row of inner hair cells and three rows of outer hair cells (*Figure 6C*, *Control*). By contrast, *Inhba* cKO cochlear tissue showed a significant increase in the number of ectopic inner hair cells compared to control, with more of the ectopic inner hair cells residing within the most basal segment (*Figure 6C*, *Inhba* cKO, *D*). Furthermore, consistent with a delayed onset of hair cell differentiation, hair cell stereocilia in the base, mid and apex of *Inhba* mutant cochlear tissue had a less mature phenotype compared to their counterparts in control cochlear tissue (*Figure 6C*). Taken together, our findings demonstrate that *Inhba* (Activin A) function is required for limiting inner hair cell formation as well as for timing hair cell differentiation/maturation in vivo.

The milder defects in cochlear hair cell formation observed in *Inhba* mutants suggest that in addition to Activin A signaling, FST likely disrupts other, Activin-related signaling mechanisms in the developing cochlea. Like Activin A, Activin B (*Inhbb*) and the Activin-related ligands GDF11 and Myostatin (*Mstn*) activate SMAD2/3 signaling and are high affinity binding partners of FST (*Thompson et al., 2005*; *Walker et al., 2017*). To determine whether *Gdf11*, *Inhbb*, *Mstn* are expressed within the developing cochlear epithelial duct, we conducted a series of RT-qPCR experiments (*Figure 6—figure supplement 1*). We found *Gdf11* transcript to be abundantly expressed at the onset of (E13.5) as well as during (E14.5, E15.5) cochlear differentiation, making it a likely candidate for FST-mediated regulation. By contrast, *Inhbb* and *Mstn* transcripts were undetectable or expressed at very low levels.

## Overexpression of FST delays cochlear pro-sensory cell cycle exit

The supernumerary inner hair cells and inner phalangeal cells observed in *R26-FST* transgenic animals could have been the product of prolonged proliferation of medially located pro-sensory cells, or they could have been created by a shift in the medial boundary of the pro-sensory domain, leading to the recruitment of non-sensory cells to develop as inner hair cell and inner phalangeal cells (*Basch et al., 2016*). To distinguish between these two possibilities, we induced FST overexpression at E11.5 and analyzed the spatial and temporal pattern of pro-sensory cell cycle exit in *R26-FST* transgenic animals and their non-transgenic littermates (control). In the murine cochlea, cell cycle exit occurs within a 48 hr time window, with apically located pro-sensory cells exiting the cell cycle as early as E12.5 and pro-sensory cells located in the base of the cochlea exiting as late as E14.5 (*Lee et al., 2006*; *Ruben, 1967*). In our first set of experiments we injected timed pregnant dams with the thymidine analog EdU at stage E13.5, which corresponds with the peak of pro-sensory cell

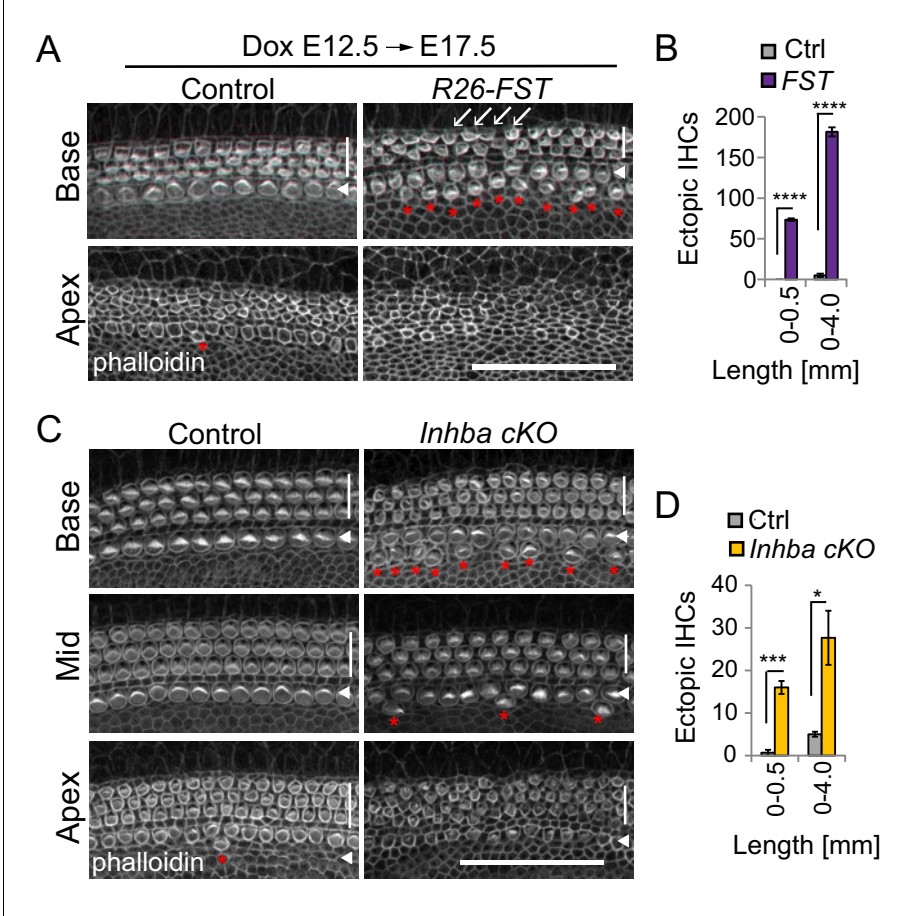

**Figure 6.** Activin signaling is required for patterning and maturation of hair cells. (A–B) FST overexpression at E12.5 delays hair cell maturation and causes a mild overproduction of inner hair cells. *FST* transgenic (*R26-FST*) embryos and their control (single transgenic) littermates were exposed to dox starting at E12.5 until tissue harvest at E17.5. (A) Shown are the apical surfaces of hair cells located in the base, mid and apex of control and *R26-FST* transgenic cochlear sensory epithelia. Phalloidin labels actin-rich stereocilia of inner (white arrowhead) and outer hair cells (white bar). Red asterisks mark ectopic inner hair cells. White arrows mark location of missing outer hair cells. Scale bar 50 µm. (B) Graphed are the number of ectopic inner hair cells (IHC) within the most basal segment (0–0.5 mm) and within the entire length (0–4 mm) of control (Ctrl, gray) and FST overexpressing (FST, purple) cochleae. Data expressed as mean ± SEM (n = 3 animals per group, ****p<0.0001, student's t-test). (C-D) Conditional ablation of the *Inhba* gene delays hair cell maturation and causes a mild overproduction of inner hair cells. (C) Shown are the apical surfaces of hair cells located in the base, mid and apex of stage P0 *Inhba fl/fl* (control) and *Pax2-Cre Inhba fl/fl* (*Inhba* cKO) cochlear sensory epithelia. Phalloidin labels actin-rich stereocilia of inner (white arrowhead) and outer hair cells (white bar). Red asterisks mark ectopic inner hair cells. Scale bar 50 µm. (D) Graphed are the number of ectopic inner hair cells (IHC) within the most basal segment (0–0.5 mm) and within the entire length (0–4 mm) of control (Ctrl, gray) and *Inhba* mutant (*Inhba* cKO, orange) cochleae. Data expressed as mean ± SEM (n = 3 animals per group, *p≤0.05, ***p<0.001).

DOI: https://doi.org/10.7554/eLife.47613.008

The following figure supplement is available for figure 6:

**Figure supplement 1.** Characterization of Activin-type ligand expression in cochlear epithelial cells.
DOI: https://doi.org/10.7554/eLife.47613.009

cycle withdrawal and analyzed EdU incorporation in differentiated auditory hair cell and supporting cells five days later (stage E18.5).

Consistent with previous reports, we found that hair cells located in the basal turn of the cochlea incorporated EdU at high rate (~25%) in both control and *R26-FST* transgenic animals (*Figure 7A–C*). However, while control cochlear hair cells located further apically (mid and apex) showed little to

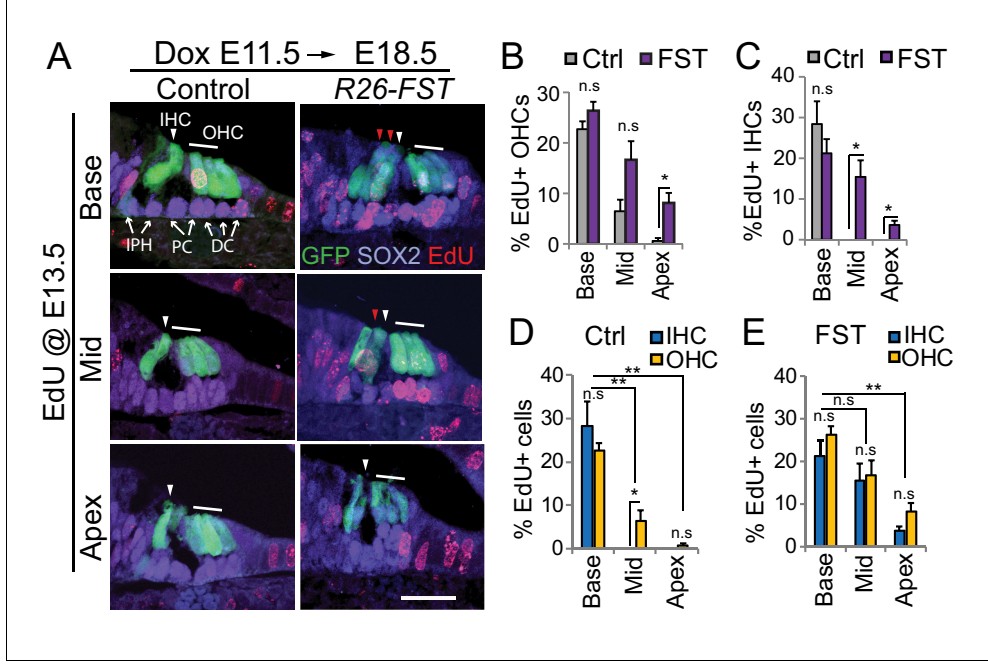

**Figure 7.** FST overexpression delays pro-sensory cell cycle exit. Timed mated pregnant dames received dox beginning at E11.5, followed by a single injection of EdU at E13.5. At E18.5, *FST* transgenic (*R26-FST*) embryos and their control (wild type or single transgenic) littermates were harvested and analyzed for EdU incorporation (red). (A) Shown are cross-sections through the base, mid and apical turn of control and FST overexpressing cochlear tissue. *Atoh1-GFP* transgene expression (GFP, green) marks inner hair cells (IHC, white arrowhead) and outer hair cells (OHC, white bar). SOX2 immunostaining (magenta) marks supporting cells including inner phalangeal cells (IPH), pillar cells (PC) and Deiters' cells (DC) marked by white arrows. Ectopic inner hair cells are marked by red arrowheads. Scale bar, 50 μm. (B–E) Graphed are the percentage of EdU positive outer hair cells (OHC) and inner hair cells (IHC) that were observed within the base, mid and apex of control (Ctrl) and FST overexpressing (FST) cochlear whole mounts. Data expressed as mean ± SEM (n = 4 animals per group, *p≤0.05, **p<0.01, student's t-test).

DOI: https://doi.org/10.7554/eLife.47613.010

no EdU incorporation, their counterparts in FST overexpressing cochleae exhibited robust EdU incorporation (*Figure 7A–C*), indicating that FST overexpression in the developing cochlea delays pro-sensory cell cycle withdrawal. FST overexpression did not disrupt the apical-to-basal gradient of pro-sensory cell cycle withdrawal. This was illustrated by the fact that similar to control tissue, inner and outer hair cells located at the apex of FST overexpressing tissue incorporated EdU a significantly lower rate than their counterparts located in the base of the cochlea (*Figure 7A,D,E*).

## Overexpression of FST abolishes a medial-to-lateral gradient of pro-sensory cell cycle exit

Interestingly, our analysis revealed that in wild type cochlear tissue inner and outer hair cell progenitors did not withdraw from the cell cycle at the same time. We found that in the mid-turn of control cochleae more than 6% of outer hair cells were labeled with EdU, whereas no EdU incorporation was found in inner hair cells, suggesting that inner hair cell progenitors had exited the cell cycle prior to the more laterally located outer hair cell progenitors (*Figure 7D*). By contrast, no difference in the frequency of EdU incorporation among inner and outer hair cells was observed in FST overexpressing cochleae (*Figure 7E*).

To further characterize the effects of FST on inner hair cell progenitor proliferation, we administered EdU daily from E14.5 until E17.5, and analyzed EdU incorporation in hair cells and supporting cells at E18.5. Again, consistent with a severe delay in pro-sensory cell cycle exit, supporting cells and hair cells in *R26-FST* transgenic animals incorporated EdU at much higher frequency than their counterparts in control (single transgenic) animals (*Figure 8A–D*). In control cochlear tissue,

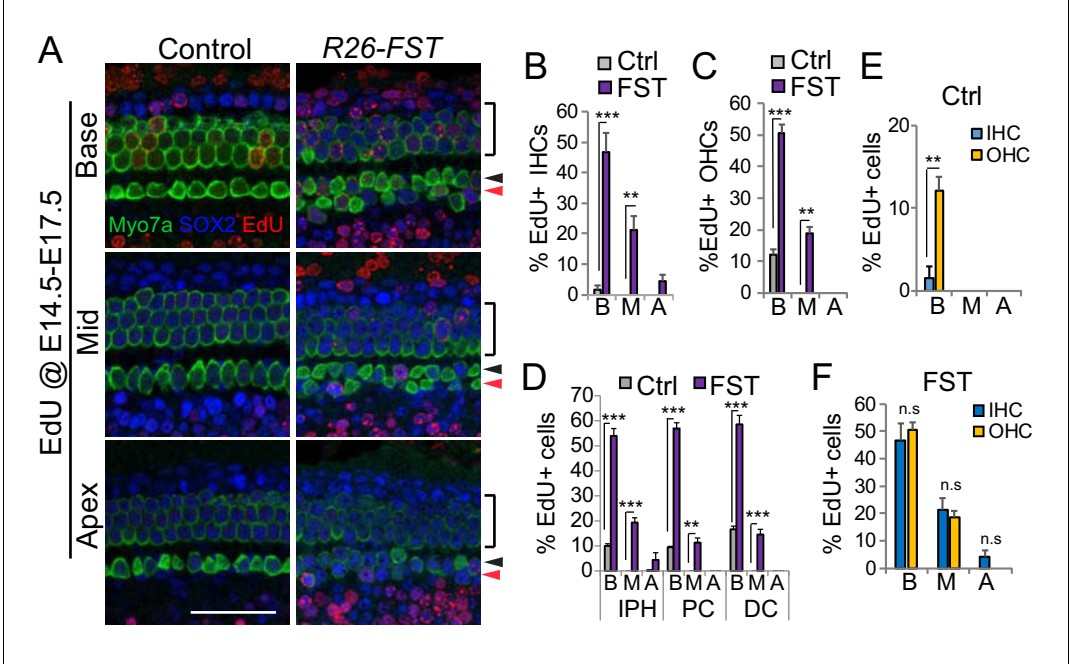

**Figure 8.** Overexpression of FST disrupts a medial-lateral gradient of pro-sensory cell cycle exit. Timed mated pregnant dam received dox beginning at E11.5, followed by two daily injections of EdU starting at E14.5 until E17.5. At E18.5, *FST* transgenic (*R26-FST*) embryos and their control (single transgenic) littermates were harvested and analyzed for EdU incorporation (red). (**A**) Shown are basal, mid and apical segments of control and FST overexpressing auditory sensory epithelia. Myo7a immuno-staining (green) marks inner (black arrowhead), outer (black bar) and ectopic inner hair cells (red arrow head). SOX2 immuno-staining (blue) marks surrounding supporting cells and less mature hair cells. Scale bar, 50 μm. (**B–C**) FST overexpression (FST, purple bars) significantly increases the percentage of EdU positive inner (IHC) (**B**) and outer hair cells (OHC) (**C**) compared to control (Ctrl, gray bars). (**D**) FST overexpression (FST, purple bars) significantly increases the percentage of EdU positive inner phalangeal cells (IPH), pillar cells (PC) and Deiters cells compared to control (Ctrl, gray bars). (**E**) In control cochlear tissue IHCs (blue) incorporate EdU at a significantly lower rate than OHCs (yellow). (**F**) In FST overexpressing cochlear tissue IHCs (blue) and OHCs (yellow) incorporate EdU at a similar rate. Abbreviations: B, base; M, mid; A, apex. Data expressed as mean ± SEM (n = 3 animals per group **p<0.01, ***p<0.001, student's t-test).

DOI: https://doi.org/10.7554/eLife.47613.011

consistent with the existence of a medial-to-lateral gradient of cell cycle withdrawal, the percentage of outer hair cells (12.1%) that incorporated EdU was significantly higher compared to the percentage of inner hair cells (1.5%) that incorporated EdU (*Figure 8E*). By contrast, in FST overexpressing cochlear tissue, inner hair cells incorporated EdU at a similar or higher rate than outer hair cells (*Figure 8F*) and in its apex inner hair cells and their surrounding inner phalangeal cells were the only cell types that were labeled with EdU (*Figure 8A,B,D*).

Taken together, these findings suggest that the ectopic hair cells observed in FST overexpressing cochlear tissue are, at least in part, the product of prolonged proliferation of inner hair cell progenitor cells.

## Discussion

Signaling gradients play a fundamental role in controlling growth and differentiation during embryonic development. Here, we identify Activin A and its antagonist FST as key regulators of auditory hair cell differentiation. We show that Activin A acts as differentiation signal for sensory hair cells and present evidence that a counter gradient of Activin A and FST times the wave of hair cell differentiation within the developing murine cochlea. Finally, we provide evidence that Activin-type

signaling regulates a previously unrecognized radial gradient of pro-sensory cell cycle withdrawal, limiting the number of inner hair cells being produced.

## Activin A functions as a pro-differentiation signal in the mammalian cochlea

We find that exogenous Activin A triggers premature hair cell differentiation in cochlear explant cultures and demonstrate that disruption of Activin A signaling due to *FST* overexpression or *Inhba* deletion results in a delay in cochlear hair cell differentiation in vivo. How does Activin signaling promote cochlear hair cell differentiation? The function of Activin signaling in cell proliferation and cell differentiation is highly dependent on cellular context. This has been attributed to the cross-talk and cross-regulation between the Activin signaling pathway and other key developmental pathways (*Luo, 2017*). One possible scenario is that Activin signaling promotes hair cell differentiation through mitigating the hair cell-repressive activity of SHH signaling. For instance, Activin signaling has been shown to promote cortical interneuron differentiation, at least in part, through the inhibition of SHH signaling (*Cambray et al., 2012*). However, we found no evidence for direct interference as neither over-activation nor loss of Activin signaling altered the expression of SHH target and effector genes *Hey1*, *Hey2* and *Fgf20* (*Benito-Gonzalez and Doetzlhofer, 2014*; *Tateya et al., 2013*).

Another possible scenario is that Activin-SMAD2/3 signaling directly trans-activates the transcription of *Atoh1*. We did observe a robust increase in *Atoh1* transcript abundance in response to 24 hr Activin treatment. However, a brief 4 hr Activin A treatment failed to significantly increase *Atoh1* transcript abundance, suggesting that Activin signaling does not activate *Atoh1* expression directly but rather through indirect mechanisms. Indeed, our findings suggest that Activin signaling positively regulates ATOH1 activity/expression through inhibiting cochlear expression of *Id3* and *Id4*. Members of the ID family of proteins interfere with the ability of bHLH transcription factors to bind DNA and trans-activate their targets genes (*Benezra et al., 1990*; *Wang and Baker, 2015*), which in the case of ATOH1 would render it incapable of upregulating its own transcription and inducing hair cell formation (*Bermingham et al., 1999*; *Helms et al., 2000*). Consistent with an inhibitory role in hair cell differentiation, previous studies found that a failure to downregulate *Id3* expression in hair cell progenitors inhibits the formation of hair cells in both auditory and vestibular sensory organs (*Jones et al., 2006*; *Kamaid et al., 2010*). The role of ID4 in inner ear development has yet to be established.

It should be noted that *Atoh1* expression and subsequent hair cell differentiation is not disrupted, but rather delayed in response to FST overexpression or *Inhba* ablation, suggesting the existence of Activin-independent mechanisms capable of activating *Atoh1* expression in pro-sensory cells. A likely candidate is the Wnt/β-catenin signaling pathway. The Wnt effector β-catenin is required for hair cell specification and has been shown to directly activate *Atoh1* transcription in neuronal progenitors (*Shi et al., 2010*; *Shi et al., 2014*). Taking these recent findings and our findings into consideration, we propose a model in which Wnt/β-catenin signaling initiates *Atoh1* expression, whereas Activin signaling functions to enhance *Atoh1* expression/activity along the longitudinal cochlear axis.

What are the signal(s)/factor(s) that set up the distinct pattern of cochlear *Inhba* and *Fst* expression? Early SHH signaling, presumably from the ventral midline, is critical for establishing regional identity in the developing cochlea, including setting up the distinct basal–apical pattern of *Inhba* and *Fst* expression. However, subsequent SHH signaling from the spiral ganglion is dispensable for establishing/maintaining *Inhba* and *Fst* expression (*Son et al., 2015*), suggesting the involvement of additional, yet to be identified upstream regulators. A potential candidate for inducing *Inhba* expression at the onset of hair cell differentiation is the Notch signaling pathway. Notch signaling is active in pro-sensory cells and has been shown to positively regulate *Inhba* expression in differentiating and terminal differentiated supporting cells (*Campbell et al., 2016*; *Maass et al., 2016*).

## FST maintains pro-sensory cells in a proliferative and undifferentiated state

We show that FST overexpression delays the onset of pro-sensory cell cycle withdrawal and differentiation. A similar function in timing pro-sensory cell withdrawal and differentiation has been recently reported for the RNA binding protein LIN28B (*Golden et al., 2015*). Interestingly, the expression of *Fst* in the developing cochlea largely mimics that of *Lin28b*; each are initially highly expressed in

pro-sensory cells and are downregulated upon differentiation following a basal-to-apical gradient. Regulatory and functional connections between Activin signaling and LIN28B have not yet been established and it will be of interest to determine whether a link between LIN28B and FST exists.

TGF-β-related signaling pathways are known to inhibit proliferation through promoting the expression and/or activity of cyclin dependent kinase (CDK) inhibitors (*Massagué and Gomis, 2006*). In the developing cochlea, the CDK inhibitor p27/Kip1 (CDKN1B) is the main regulator of pro-sensory cell cycle exit and its transcriptional upregulation directs the apical-to-basal wave of pro-sensory cell cycle withdrawal (*Chen and Segil, 1999*; *Lee et al., 2006*). However, we observed no changes in p27/Kip1 mRNA expression in response to FST overexpression, suggesting that the observed delay in pro-sensory cell cycle exit may instead be due to a reduction in p27/Kip1 protein stability and/or activity. In addition, it is likely that the increase in *Id3* and *Id4* expression in response to FST overexpression contributed to the observed delay in pro-sensory cell cycle exit. High ID protein expression is associated with a proliferative, undifferentiated cell state and ID4 expression in spermatogonia and mammary glands is a predictor for stemness (*Helsel et al., 2017*; *Wang and Baker, 2015*). Does FST target other Activin-type ligands in the developing cochlea? This is quite likely as *Fst* can be detected within the most ventral aspect of the developing otocyst as early as E10.5, preceding Activin A expression by several days (*Son et al., 2012*). Furthermore, the cochlear defects observed in *Inhba* mutants are much milder compared to the cochlear defects observed in *FST* transgenic animals. A potential candidate is GDF11, which our qPCR-based expression analysis identified as the predominant Activin-type ligand within the developing cochlea. GDF11–FST axis has been shown to regulate progenitor cell proliferation and differentiation in various sensory tissues (*Beites et al., 2009*; *Gokoffski et al., 2011*; *Kim et al., 2005*). However, the role of GDF11 in the developing cochlea has not yet been addressed and future investigations are warranted.

Interestingly, the negative role of Activin-type signaling in cochlear cell proliferation appears to be not conserved between mammalian and avian species. In adult chicken auditory sensory epithelia disruption of Activin signaling attenuates the low rate of regenerative cell proliferation, whereas exogenous Activin A stimulates cell proliferation (*McCullar et al., 2010*).

## A radial Activin-FST counter gradient controls the production of inner hair cells

We find that FST overexpression and to a lesser extent ablation of Activin A (*Inhba*) results in an overproduction of inner hair cells. In certain tissues, including the developing tongue, FST has been shown to interfere with BMP signaling (*Beites et al., 2009*). In the mammalian cochlea BMP-receptor signaling is critical for the specification of sensory and non-sensory territories and the medial-lateral partitioning of the pro-sensory domain (*Ohyama et al., 2010*) (*Munnamalai and Fekete, 2016*; *Puligilla et al., 2007*). Thus, it could be reasoned that the observed increase in inner hair cells in FST overexpressing cochleae is the result of defects in medial-lateral patterning caused by the loss of BMP signaling. However, we found no evidence that FST overexpression disrupted or significantly reduced BMP signaling within the developing cochlea. FST overexpression did not reduce protein levels of the BMP effectors p-SMAD1/5/9, nor did it reduce the expression of BMP target genes *Id1-3*.

Instead, we propose that the ectopic inner hair cells in *FST* transgenic and *Inhba* mutant animals are the product of prolonged inner hair cell progenitor proliferation. We show that under normal conditions inner hair cell progenitors withdraw from the cell cycle prior to outer hair cell progenitor cells. Moreover, we demonstrate that this distinct medial-to-lateral gradient of terminal mitosis is lost in FST overexpressing cochlear tissue, resulting in a disproportional increase in inner hair cells. What informs the radial gradient of terminal mitosis in wild type cochlear tissue? We show that at the peak of terminal mitosis (E13.5), *Fst* expression becomes restricted to the lateral portion of the pro-sensory domain, suggesting that inner hair cell progenitors experience higher levels of Activin-type signaling (Activin A, GDF11) than more laterally located outer hair cell progenitor cells. This newly identified radial gradient of terminal mitosis constitutes a novel mechanism for limiting the number of inner hair cells that are being produced in the mammalian cochlea.

## Materials and methods

### Experimental animals

*Atoh1-GFP* transgenic mice (RRID:MGI:3703598, Tg(Atoh1-GFP)1Jejo (*Lumpkin et al., 2003*) were obtained from Jane Johnson, University of Texas, Southwestern Medical Center. *Pax2-Cre* BAC transgenic mice (RRID:MGI:3046196, Tg(Pax2-cre)1Akg) (*Ohyama and Groves, 2004*) were obtained from Andrew Groves, Baylor College. *Inhba floxed* mice (RRID: MGI:3758877, Inhba^{tm3Zuk}) (*Pangas et al., 2007*) were obtained from Martin Matzuk, Baylor College. The *R26-M2rtTA* mice (RRID:MGI:3798943, B6.Cg-Gt(ROSA)26Sortm1(rtTA*M2)Jae/J) (*Hochedlinger et al., 2005*) were purchased from Jackson Laboratories (Bar Harbor, ME, no. 006965). *FST* transgenic mice were obtained from Se-Jin Lee, Johns Hopkins University School of Medicine. In this line a cassette encoding the human FST-288 isoform is under the control of a tetracycline-responsive promoter element (tetO) (*Lee, 2007*; *Roby et al., 2012*). Mice were genotyped by PCR using the listed primers. *Pax2-Cre*: Cre1F (GCC TGC ATT ACC GGT CGA TGC AAC GA), Cre1R (GTG GCA GAT GGC GCG GCA ACA CCA TT) yields a 700 bp band. *Inhba floxed*: Inhba fx1 (AAG AGA GAA TGG TGT ACC TTC ATT), Inhba fx2 (TAT AAC CTG GGT AAG TGG GT), Inhba fx3 (AGA CGT GCT ACT TCC ATT TG) yield a 400 bp band for the floxed allele and a 280 bp for the wild type allele. *R26-M2rtTA*: MTR (GCG AAG AGT TTG TCC TCA ACC), F (AAA GTC GCT CTG AGT TGT TAT), WTR (GGA GCG GGA GAA ATG GAT ATG) yield a 340 bp band for the mutant allele and a 650 bp for the wild type allele. *FST:* YA88 (TTG CCT CCT GCT GCT GCT GC), YA123 (TTT TTC CCA GGT CCA CAG TCC ACG) yields a 247 bp band for the *FST* transgene. *Atoh1-GFP*: EGFP1 (CGA AGG CTA CGT CCA GGA GCG CAC), EGFP2 (GCA CGG GGC CGT CGC CGA TGG GGG TGT) yields a 300 bp band for GFP. Mice were maintained on a C57BL/6; CD-1 mixed background. Mice of both sexes were used in this study. Embryonic development was considered as E0.5 on the day a mating plug was observed. To induce *FST* transgene expression doxycycline (dox) was delivered to time-mated females via ad libitum access to feed containing 2 grams of dox per kilogram feed (Bioserv, no. F2893). All experiments and procedures were approved by the Johns Hopkins University Institutional Animal Care and Use Committee (protocol #MO17M318), and all experiments and procedures adhered to the National Institutes of Health-approved standards.

### Tissue harvest and processing

Embryos and early postnatal pups were staged using the EMAP eMouse Atlas Project (http://www.emouseatlas.org) Theiler staging criteria. Inner ear cochleae were collected in HBSS (Corning, no. 21–023-CV). To free the cochlear epithelial duct from surrounding tissue, dispase II (1 mg/ml; Gibco, no. 17105041) and collagenase (1 mg/ml; Worthington, no. LS004214) mediated digest was used as previously described (*Golden et al., 2015*). To obtain cochlear whole mount preparations (also referred to as surface preparations), the cochlear capsule, spiral ganglion, and Reissner's membrane were removed, and the remaining tissue was fixed overnight in 4% (vol/vol) paraformaldehyde (PFA) (Electron Microscopy Sciences, no. 15714) in PBS. To obtain cochlear sections, heads were fixed in 4% PFA in PBS, cryoprotected using 30% sucrose in PBS, and embedded in Tissue-Tek O.C.T. Compound (Sakura Finetek). Tissue was sectioned at a thickness of 14 μm and collected on SuperFrost Plus slides (Thermo Scientific, no. 12-550-15) and stored at −80°C.

### Histochemistry and in situ hybridization

Immunostaining was performed according to the manufacturer's specifications. Primary antibodies: rabbit anti-myosinVIIa (1:500; Proteus, no. 25–6790, RRID: AB_10015251), rabbit anti SOX2 (1:500; Millipore, no. AB5603, RRID:AB_2286686) goat anti-SOX2 (1:500; Santa Cruz, no. sc-17320, RRID: AB_2286684), rabbit anti-PROX1 (1:500; Covance, no. PRB-238C, RRID: AB_291595). Cell nuclei were fluorescently labeled with Hoechst-33258 solution (Sigma, no. 94403). Actin filaments were labeled with Alexa Fluor 488 phalloidin (1:1000; Invitrogen, no. A12379) or Alexa Fluor 546 phalloidin (1:1000; Invitrogen, no. A22283). Secondary antibodies: anti-rabbit and anti-goat IgG (H + L) Alexa Fluor (488, 546 and 647) labeled secondary antibodies (1:1000; Invitrogen) and biotinylated donkey anti-goat IgG (H + L) (1: 500; Jackson Immuno Research laboratories, no. 705-065-147) with streptavidin, Alexa Fluor 405 (1:100; Invitrogen, no. S32351) were used for immunostaining. For in situ hybridization digoxigenin (DIG)-labeled antisense RNA probes were prepared according to

the manufacturer's specifications (Roche). PCR amplified fragments of *Inhba* (NM_008380, 47–472) and *Fst* (NM_008046, 180–492) were used as template and gene-specific T7 RNA polymerase promoter hybrid primers were used for in vitro transcription. *Atoh1* (NM_007500) and *Sox2* (NM_011443) probes were prepared as previously described (Golden et al., 2015). Probes were detected with the sheep-anti–DIG-alkaline phosphatase (AP) conjugated antibody (1:000; Roche, no. 11093274910) and the color reactions were developed by using BM Purple AP Substrate (Roche, no. 11442074001).

## Recombinant protein
Recombinant human/mouse/rat Activin A (R&D Systems, no. 338-AC) was reconstituted in sterile PBS containing 0.1% BSA at a concentration of 50 μg/ml and used at final concentration of 200–500 ng/ml. Recombinant human BMP4 (R&D Systems, no. 314 BP-010) was reconstituted in sterile 4 mM HCl 0.1% BSA at a concentration of 100 μg/ml and used at 100 ng/ml final concentration. Stock solutions were stored at −20°C.

## Cochlear explant culture and hair cell differentiation assay
Wild type or FST overexpressing embryos and their control littermates were screened for native GFP expression and staged (see tissue harvest and processing). Embryos of inappropriate stage and GFP negative embryos were discarded. Cochleae from individual embryos were harvested in HBSS (Corning, no. 21–023-CV), and treated with dispase and collagenase to remove the cochlear capsule (see tissue harvest and processing). The remaining tissue, including the cochlear epithelial duct, the vestibular sacculus, and the innervating spiral ganglion, was placed onto SPI-Pore membrane filters (Structure Probe, no. E1013-MB) and cultured in DMEM-F12 (Corning, no. 10–092-CV), 1% Fetal Bovine Serum (FBS) (Atlanta Biologicals), 5 ng/ml EGF (Sigma, no. SRP3196), 100 U/ml penicillin-streptomycin (Sigma, no. P4333), and 1x B-27 Supplement (Gibco, no. 17504044). Activin A (final conc. 500 ng/ml) was added at plating and was replenished daily. All cultures were maintained in a 5% $CO_2$/20% $O_2$ humidified incubator. To monitor hair cell differentiation, green fluorescent images of native GFP expression were captured using fluorescent stereo-microscopy (Leica, Microsystems). Fluorescent images were analyzed in Photoshop CS6 (Adobe), and the basal-to-apical extent of cochlear GFP expression was measured using ImageJ software (National Institutes of Health).

## RNA extraction and RT q-PCR
Cochlear epithelia were isolated from cultured cochlear explants or freshly harvested inner ear tissue using dispase/collagenase treatment. RNeasy Micro kit (Qiagen, no.74004) was used to isolate total RNA, and mRNA was reverse transcribed into cDNA using iScript kit (Bio-Rad, no. 1708890). Q-PCR was performed with Fast SYBR Green Master Mix reagent (Applied Biosystems, no. 4385614) and gene-specific primer sets on a CFX-Connect Real Time PCR Detection System (Bio-Rad). Each PCR was performed in triplicate. Relative gene expression was analyzed by using the ΔΔCT method (Schmittgen and Livak, 2008). The ribosomal gene *Rpl19* was used as endogenous reference gene. The following q-PCR primers were used:

| Gene | Forward Primer | Reverse Primer |
| --- | --- | --- |
| *Atoh1* | GGCAACAGCTCCCTGAAAACT | CCCGCGCGCTAGGAA |
| *Bmp4* | ACGTAGTCCCAAGCATCACC | ACTAGGGTCTGCACAATGGC |
| *Etv5* | GAGCCGCTCTCTCCGCTATT | CGTTCCCCAGCCACCTT |
| *FST* (h) | TTGCCTCCTGCTGCTGCTGC | CTCCTTGCTCAGTTCGGTCTTG |
| *Fst* (m) | GCTCTTCTGGCGTGCTTC | AATGGAAGAGATAGGAAAGCTGT |
| *Fgf20* | CACGGGTCGCAGGTATTTTG | CCTGGCACCATCTCTTGGA |
| *Gdf11* | GCGTCACATCCGTATCCGTT | TCCATGAAAGGATGCAGCCCC |
| *Hey1* | CACTGCAGGAGGGAAAGGTTAT | CCCCAAACTCCGATAGTCCAT |
| *Hey2* | AAGCGCCCTTGTGAGGAAA | TCGCTCCCCACGTCGAT |

*Continued on next page*

*Continued*

| Gene | Forward Primer | Reverse Primer |
| --- | --- | --- |
| *Id1* | GAACGTCCTGCTCTACGACATG | TGGGCACCAGCTCCTTGA |
| *Id2* | AAGGTGACCAAGATGGAAATCCT | CGATCTGCAGGTCCAAGATGT |
| *Id3* | *GAGCTCACTCCGGAACTTGTG* | CGGGTCAGTGGCAAAAGC |
| *Id4* | TGCGATATGAACGACTGCTACA | TTGTTGGGCGGGATGGTA |
| *Inha* | GCTCCTTTTGCTGTTGACCC | GTGGCACCTGTAGCTGGGAA |
| *Inhba* | GGGTAAAGTGGGGGAGAACG | ACTTCTGCACGCTCCACTAC |
| *Inhbb* | GCGTCTCCGAGATCATCAGC | CACCTTGACCCGTACCTTCC |
| *Lgr5* | TGAGCGGGACCTTGAAGATTT | AATAGGTGCTCACAGGGCTT |
| *Mstn* | TCTTGCTGTAACCTTCCCAGG | TCGCAGTCAAGCCCAAAGTC |
| P27(*Cdkn1b*) | GCAGGAGAGCCAGGATGTCA | CCTGGACACTGCTCCGCTAA |
| *Pou4f3* | GCACCATCTGCAGGTTCGA | CCGGCTTGAGAGCGATCAT |
| *Ptch1* | CTGGCTCTGATGACCGTTGA | GCACTCAGCTTGATCCCAATG |
| *Rpl19* | GGTCTGGTTGGATCCCAATG | CCCGGGAATGGACAGTCA |

## Western blot

Individual cochlear epithelia were lyzed in RIPA lysis buffer (Sigma, no. R0278) supplemented with Roche Protease Inhibitor (Sigma, no. 11697498001) and Phosphatase Inhibitor Cocktail no.2 (Sigma, no. P5726) and no.3 (Sigma, no. P0044). Following manufacture recommendations, equal amounts of cochlear protein extract were resolved on NuPAGE 4–12% Bis-Tris Gels (Invitrogen, no. NP0322BOX) and transferred to Immun-Blot PVDF membrane (Bio-Rad) by electrophoresis. Membranes were blocked in 5% no-fat dry milk in TBST and immunoblotted with rabbit anti-P-Smad2/3 (1:1000; Cell Signaling, no.8828 RRID: AB_2631089), P-Smad1/5/9 (1:1000; Cell Signaling, no.13820, RRID: AB_2493181) and mouse anti-β-actin (1:1000; Santa Cruz, no. SC-47778, RRID: AB_626632). HRP-conjugated secondary antibodies from Jackson Immuno Research were used at a concentration of 1:10,000 (goat anti-rabbit IgG, no. 111-035-003; sheep anti-mouse IgG no. 515-035-003). Signal was revealed using a Western Lightening Plus ECL kit (Perkin Elmer, no. NEL120E001EA) or Super-Signal West Femto Maximum Sensitivity Substrate (Thermo Scientific, no. 34096) according to manufacturer's instructions.

## Quantification of hair cells and supporting cells

Cell counts were performed in cochlear whole mounts. Hair cells were identified by their native GFP (*Atoh1-GFP*) expression, as well as by immuno-staining for the hair cell-specific protein myosinVIIa (Myo7a); supporting cells were identified by SOX2 immuno-staining. Classification of cell subtypes was based on their location within the sensory epithelium. Low-power confocal and epi-fluorescent images (ZEISS, Microscopy) of the hair cell layer were used to reconstruct the entire cochlear sensory epithelium. The resulting composite images were used to count ectopic inner hair cells, measure the total length of the sensory epithelia and used to define basal, mid and apical segments (~1400 μm). For hair cell and supporting cell counts a series of high-power confocal (ZEISS, Microscopy) z-stack images spanning the hair cell and supporting cell layer were taken within the basal, mid and apical segments. The apical tip (~300–500 μm) was excluded from the analysis. Images were assembled, viewed and analyzed in Photoshop CS6 (Adobe) and LSM image viewer software (ZEISS, Microscopy). Image J (National Institutes of Health) and FIJI software were used to measure the length of counted segments and total length of the sensory epithelium. The FIJI cell counter plugin was used to count cells.

## Proliferation assay

EdU (5-ethynyl-2'-deoxyuridine, Invitrogen no. E10187) was reconstituted in PBS and administered at 50 μg per gram of body weight to time-mated pregnant dams per intraperitoneal injection. Click-iT AlexaFluor-488 or -594 Kit (Invitrogen, no. C10337 and no. C10339) was used to detect

incorporated EdU according to the manufacturer's specifications. To quantify the number and per-centage of EdU positive hair cells and supporting cells, EdU stained cochlear surface preparations were immuno-stained for SOX2 and Myo7a. Length measurements and cell counts were conducted as described above.

## Statistical reporting

Values are presented as mean ± standard error of the mean (SEM). The sample size (n) represents the number of biological independent samples (biological replicates) analyzed per experimental group. Two-tailed unpaired Student's t-tests were used to determine the confidence interval, p-val-ues≤0.05 were considered significant. p-values>0.05 were considered not significant. Biologically independent samples (biological replicates) were allocated into experimental groups based on genotype and/or type of treatment. A minimum of three biological independent samples were ana-lyzed per group. To avoid bias masking was used during data analysis.

## Acknowledgements

We thank the members of the Doetzlhofer Laboratory for the help and advice provided throughout the course of this study. We thank Jane Johnson for the *Atoh1-GFP* transgenic mice; Andrew Groves for the *Pax2-Cre* transgenic mice; Se-Jin Lee for *FST* transgenic mice and Martin Matzuk for *Inhba* floxed mice.

## Additional information

### Funding

| Funder | Grant reference number | Author |
| --- | --- | --- |
| National Institute on Deafness and Other Communication Disorders | DC011571 | Angelika Doetzlhofer |
| National Institute on Deafness and Other Communication Disorders | DC013477 | Ana Benito-Gonzalez |
| National Institute on Deafness and Other Communication Disorders | DC012972 | Erin Jennifer Golden |
| National Institute on Deafness and Other Communication Disorders | DC016538 | Meenakshi Prajapati-DiNubila |
| National Institute on Deafness and Other Communication Disorders | DC005211 | Angelika Doetzlhofer |
| David M. Rubenstein | Fund for Hearing Research | Angelika Doetzlhofer |

The funders had no role in study design, data collection and interpretation, or the decision to submit the work for publication.

### Author contributions

Meenakshi Prajapati-DiNubila, Formal analysis, Validation, Investigation, Writing—review and editing; Ana Benito-Gonzalez, Conceptualization, Formal analysis, Investigation, Writing—original draft, Writing—review and editing; Erin Jennifer Golden, Investigation, Methodology, Writing—review and editing; Shuran Zhang, Investigation, Methodology; Angelika Doetzlhofer, Conceptualization, Data curation, Formal analysis, Supervision, Funding acquisition, Writing—original draft, Writing—review and editing

## Author ORCIDs

Meenakshi Prajapati-DiNubila [ID] http://orcid.org/0000-0002-4600-9778
Angelika Doetzlhofer [ID] https://orcid.org/0000-0002-7424-2112

## Ethics

Animal experimentation: This study was performed in strict accordance with the recommendations in the Guide for the Care and Use of Laboratory Animals of the National Institutes of Health. All experiments and procedures were approved by the Johns Hopkins University Institutional Animal Care and Use Committee (IACUC), protocol #MO17M318.

## Decision letter and Author response

Decision letter https://doi.org/10.7554/eLife.47613.014
Author response https://doi.org/10.7554/eLife.47613.015

## Additional files

### Supplementary files

• Transparent reporting form
DOI: https://doi.org/10.7554/eLife.47613.012

### Data availability

All data generated or analysed during this study are included in the manuscript and supporting files.

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
