## [Decision Letter]

[Editors’ note: a previous version of this study was rejected after peer review, but the authors submitted for reconsideration. The first decision letter after peer review is shown below.]

Thank you for choosing to send your work entitled "Activin signaling informs the graded pattern of terminal mitosis and hair cell differentiation in the mammalian cochlea" for consideration at *eLife*. Your initial submission has been assessed by a Senior Editor in consultation with a member of the Board of Reviewing Editors. Although the work is of interest, we are not convinced that the findings presented have the potential significance that we require for publication in *eLife*.

Specifically, while the reviewers agreed that the results are potentially significant, they felt that many modifications and new experiments would be required to bring the paper to a standard that would be acceptable for *eLife*. We refer you to their specific reviews below. Important considerations include discussion of other pathways that may regulate timing of proliferation and differentiation in the cochlea, better citing of the existing literature and further experimentation. In particular, the reviewers were sceptical about some of the crucial interpretations, and felt these needed to be supported by stronger experimental results. If you feel you can address these issues in a future publication, we would be open to considering a significantly revised paper as a new submission for which every effort would be made to return the paper to the same reviewers.

*Reviewer #1:*

This is an interesting and thorough study that presents new data illustrating a role for Activin/Follistatin signaling in the timing of hair cell differentiation and cellular proliferation within the cochlea. The data are nicely presented and the overall conclusions, that counter gradients of Activin A and Follistatin regulate the timing of hair cell differentiation along the cochlear axis are well supported. My largest concern with the study is that while discussing three other signaling pathways, Shh, FGF and Wnt, that have been shown to also play roles in regulating hair cell differentiation, meaningful data exploring the links between Activin A/Follistatin and any of these other pathways is not presented. Some possibilities are broached in the Discussion section, but no specific interactions are examined. In particular, additional exploration of the link between the Shh/*Fgf20* pathway, which the authors showed to be important for the gradient of hair cell differentiation in a previous study and Activin A/Follistatin, would greatly strengthen this work.

*Reviewer #2:*

The paper by Prajapati-DiNubila et al., reports the important observation that Activin A and follistatin play a role in the timing of onset of differentiation of the sensory hair cells in the developing organ of Corti. This question; "What is responsible for timing the onset and progression of HC differentiation along the basal-to-apical gradient?" is an extremely important one, and the identification of a role for Activin in this process may answer this long-standing issue. The paper clearly shows the effects of Activin and follistatin, but there are a fairly large number procedural and control issues to be dealt with, as well as questions about interpretation.

Using in situ hybridization, in Figure 1, the authors first replicate the observation that Activin A follows a basal-to-apical gradient of expression within the sensory domain of the organ that parallels the timing of hair cell differentiation (Figure 1). They also report a medial-lateral gradient of Activin A and follistatin, that they claim to observe by in situ hybridization, but this is harder to discern in the data as presented (see comment below).

Using an ex vivo culture system that allows hair cell differentiation to proceed from base-to-apex starting at a time just before the appearance of Atoh1, a crucial regulator of hair cell differentiation, the authors show that incubation of the cultures with Activin A leads to a precocious differentiation process, as measured by the basal-to-apical extent of transgenic GFP, reporting Atoh1 protein level, in differentiating hair cells (Figure 2A-G).

Interestingly, but perhaps inconsistent with the authors claim that Activin A regulates Atoh1 directly, they observe that once initiated, the process (rate) of differentiation along the basal-to-apical axis is the same as in the untreated cultures, and patterning is unaffected. Since the whole organ is bathed in exogenous Activin A at the beginning of the experiment, one might expect that Atoh1 would be up-regulated in the apex, as well as the base, however this does not seem to occur. Also, that Activin might stimulate Atoh1 in nascent supporting cells inducing transdifferentiation, which also does not seem to occur. This suggests that it is unlikely that Activin A is regulating Atoh1 directly, and that the modest Atoh1 increase shown by qPCR may represent the change in timing of the initiation of the wave of differentiation.

The Activin A effect on Atoh1 up-regulation is shown to be specific by comparison to BMP4, by qPCR (Figure 2K), though the effect is rather small. Here the data are confounded by use of a different experimental approach, in which a different dissection eliminated surrounding tissue. (subsection “Activin signaling positively regulates Atoh1 gene expression” and Figure 2K) "We shortened the time of Activin A treatment to 3.5-hours, as well as cultured cochlear epithelia without its surrounding tissue." This requires some explanation. What is meant by "surrounding tissue" and was this included in the previous qPCR experiment (Figure 2I), or not? It seems that these details of the culture conditions could influence the outcome of the experiment, and so defining them precisely is important.

Activin activity is known to be opposed by follistatin, and in Figure 3, the authors make the case that follistatin limits the differentiation of hair cells. Overexpression of follistatin inhibits the normal basal-to-apical progression of HC differentiation that occurs between E13.5 and E17.5. Together these results suggest a role for ActivinA in timing this progression and a reliance on Folistatin to oppose this process, though why both need to be regulated is not clear. At the gene expression level, it is unclear from the figure whether the change claimed for Atoh1 or Id4 is statistically significant Figure 3F, though the change in phosphor-Smad2/3 in response to follistatin seem to be (Figure 3E). However, the data in Figure 3E needs to be quantified, and the total SMAD 2/3 protein levels shown using a non-phospho-specific antibody. The ability of follistatin overexpression to antagonize the effect of Activin A is shown in Figure 3H-T. Here it would be helpful to better define the "hair cell domain" measurement used.

Interestingly, the authors show that Follistatin overexpression appears to delay cell cycle exit in an apical-to-basal direction. And also, claim to show a normal medial-to-lateral gradient of cell cycle exit that is also opposed by follistatin overexpression. However, the data presented in Figure 4I is very difficult to understand, perhaps because of the way it is graphed. It might be useful for the authors to show these data from IHCs and OHCs on two separate graphs (with associated statistics). Nonetheless, the numbers seem to indicate that follistatin increases the percentage of OHCs that take up EdU in the mid-cochlea relative to IHC, suggesting differential sensitivity from medial-to-lateral.

However, this result appears to contrast with the result in the earlier part of Figure 4, which shows the ectopic proliferation in the IHC region to be more robust? Whereas, in Figure 5, where there appears to be an increase in ectopic inner HCs, accompanied by either normal or depleted three rows of outer hair cells in FST-OE. I would have expected, based on Figure 4K, that the opposite might be true

Finally, in Figure 5, the authors use an Inhba knockout strategy, and discover that there are sporadic ectopic inner HCs present in the KO, with an increase from base-to-apex. However, given the increase in proliferation seen in the follistatin OE, one might expect both IHC and OHC to be affected, and that the effect would be stronger and less organized than that shown in Figure 5Q-S. The phenotype shown, that of ectopic IHC occurring in a single row, seems to indicate a problem with border formation, rather than an overproliferation phenotype?

*Reviewer #3:*

The manuscript "Acitivin signaling informs the graded pattern of terminal mitosis and hair cell differentiation in the mammalian cochlea" by Prajapati-DiNublia et al., examines the role of Activin A in hair cell differentiation. Specifically, using cochlear explant cultures as well as genetic mouse models the authors provide evidence that Acivin A is a positive regulator of hair cell differentiation along the basal-to-apical axis of the cochlear duct. Further evidence is provided for a role of Acivin A signaling in terminal mitosis along the medial-to-lateral axis of the cochlear sensory epithelium.

Essential revisions:

In McCullar et al., 2010, the study demonstrated that the addition of Activin A to post-hatch avian auditory sensory epithelium increases proliferation while inhibiting the Activin receptors Acvr2a and 2b results in a decrease in proliferation in the sensory epithelium. This significantly lessens the novelty/impact/advance of the current study. It is unfortunate that the authors did not reference this paper.

Furthermore, in the 2010 study McCullar et al., also demonstrated the expression patterns of the Activin receptors Acvr2a and 2b as well as the downstream Smad effector proteins pSmad2 and pSmad1/5/8 in the mammalian cochlea and vestibule (Figure 11, Figure 12, Figure 13 and Figure 14 and Table 4).

The expression patterns presented in Figure 1 have been previously published as well. Specifically, the expression of Inhba and Fst have been published by Son et al., 2015 at E13.5 (Figure S5), E15.5 (Figure 5) and E18.5 (Figure S6) clearly demonstrating the spatiotemporal basal-to-apical expression of these two molecules. The Atoh1 and *Sox2* expression patterns have also been previously published by many groups including the current authors. Thus, Figure 1 does not offer any significant new information.

[Editors’ note: what now follows is the decision letter after the authors submitted for further consideration.]

Thank you for resubmitting your work entitled "Counter gradients of Activin A and follistatin instruct the timing of hair cell differentiation in the mammalian cochlea" for further consideration at *eLife*. Your revised article has been favourably evaluated by Marianne Bronner (Senior Editor), Tanya Whitfield (Reviewing Editor), and two reviewers.

The authors have done a very nice job of addressing the reviewers' comments. There are a few remaining suggestions from the reviewers. The further experimental work suggested by reviewer 1 is left to the authors' discretion. The minor comments from reviewer 2 should be addressed by toning down the interpretations or conclusions in the manuscript.

*Reviewer #1:*

This is a re-submission by Prajapati-DiNublia et al., examining the role of Activin A in hair cell differentiation.

Initial review:

The expression patterns presented in Figure 1 have been previously published as well. […] Thus, Figure 1 does not offer any significant new information.

Authors' response:

The purpose of Figure1 is to characterize Atoh1, Inhba and Fst expression at the onset of hair cell differentiation, which the aforementioned study did not provide.

Second review:

I still fail to see what is new in Figure 1. I do appreciate it as an introductory figure.

Initial review:

In Figure 5, it would be best to have low magnification images as well to show the entire cochlea from control and mutant mice in addition to the high mag images. Are the lengths of the cochleae different – measurements? Are the extra hair cells functional? Are they innervated?

Authors response:

We quantified the length of Inhba mutant and control cochlear tissue and found no difference; the quantitative data are included in the result section. The question whether these extra hair cells are functional is interesting but beyond the scope of our current study.

Second review:

I was not suggesting electrophysiological experiments which I agree would be beyond the scope of this study; rather FM1-43 uptake which would provide evidence of channel/stereociliary function. Furthermore, performing immunohistochemistry to demonstrate innervation phenotypes (if any) is straight forward and would add value to the study.

Lastly, I do agree with the other reviewer that quantification of P-SMAD should include controls showing total receptor protein using antibodies against the un-phosphorylated SMAD form. This would control for changes in total protein levels.

---

## [Author Response]

[Editors’ note: the author responses to the first round of peer review follow.]

Specifically, while the reviewers agreed that the results are potentially significant, they felt that many modifications and new experiments would be required to bring the paper to a standard that would be acceptable for eLife. […] If you feel you can address these issues in a future publication, we would be open to considering a significantly revised paper as a new submission for which every effort would be made to return the paper to the same reviewers.Reviewer #1:[…] My largest concern with the study is that while discussing three other signaling pathways, Shh, FGF and Wnt, that have been shown to also play roles in regulating hair cell differentiation, meaningful data exploring the links between Activin A/Follistatin and any of these other pathways is not presented. Some possibilities are broached in the Discussion section, but no specific interactions are examined. In particular, additional exploration of the link between the Shh/Fgf20 pathway, which the authors showed to be important for the gradient of hair cell differentiation in a previous study and Activin A/Follistatin, would greatly strengthen this work.

Additional experiments were conducted to analyze whether Activin A and FST influences Shh/*Fgf20* pathway. Our analysis revealed that Activin A treatment or FST overexpression had no significant effect on the expression of transcriptional targets of SHH/*Fgf20* pathway (Hey1, Hey2 and *Fgf20*), suggesting that there is no direct link between SHH signaling and Activin signaling.

Reviewer #2:[…] Using an ex vivo culture system that allows hair cell differentiation to proceed from base-to-apex starting at a time just before the appearance of Atoh1, a crucial regulator of hair cell differentiation, the authors show that incubation of the cultures with Activin A leads to a precocious differentiation process, as measured by the basal-to-apical extent of transgenic GFP, reporting Atoh1 protein level, in differentiating hair cells (Figure 2A-G). […] Also, that Activin might stimulate Atoh1 in nascent supporting cells inducing transdifferentiation, which also does not seem to occur.

The preservation of the basal-to-apical gradient is consistent with our observation that the expression of key regulatory genes controlled by SHH signaling are unchanged in response to Activin A treatment, indicating that Activin signaling does not interfere with SHH signaling, which has been shown to be sufficient to maintain apical progenitors undifferentiated.

This suggests that it is unlikely that Activin A is regulating Atoh1 directly, and that the modest Atoh1 increase shown by qPCR may represent the change in timing of the initiation of the wave of differentiation.

We agree with the reviewer that our current data are most consistent with Activin signaling promoting Atoh1 expression through an indirect mechanism.

The Activin A effect on Atoh1 up-regulation is shown to be specific by comparison to BMP4, by qPCR (Figure 2K), though the effect is rather small. Here the data are confounded by use of a different experimental approach, in which a different dissection eliminated surrounding tissue. (subsection “Activin signaling positively regulates Atoh1 gene expression” and Figure 2K) "We shortened the time of Activin A treatment to 3.5-hours, as well as cultured cochlear epithelia without its surrounding tissue." This requires some explanation. What is meant by "surrounding tissue" and was this included in the previous qPCR experiment (Figure 2I), or not? It seems that these details of the culture conditions could influence the outcome of the experiment, and so defining them precisely is important.

We rewrote the section that describes the experimental approach and now state that enzymatically purified cochlear epithelia were cultured.

Activin activity is known to be opposed by follistatin, and in Figure 3, the authors make the case that follistatin limits the differentiation of hair cells. Overexpression of follistatin inhibits the normal basal-to-apical progression of HC differentiation that occurs between E13.5 and E17.5. Together these results suggest a role for ActivinA in timing this progression and a reliance on Folistatin to oppose this process, though why both need to be regulated is not clear.

The utility of having both a source (Activin A) and a sink (follistatin) in signaling gradients is well documented (e.g. wnt and wnt antagonists in rostral-caudal patterning of the neural plate).

At the gene expression level, it is unclear from the figure whether the change claimed for Atoh1 or Id4 is statistically significant Figure 3F, though the change in phosphor-Smad2/3 in response to follistatin seem to be (Figure 3E). However, the data in Figure 3E needs to be quantified, and the total SMAD 2/3 protein levels shown using a non-phospho-specific antibody. The ability of follistatin overexpression to antagonize the effect of Activin A is shown in Figure 3H-T. Here it would be helpful to better define the "hair cell domain" measurement used.

We now include detailed information about how the hair cell domain measurements were conducted in the Materials and methods section. We also conducted a new set of experiments (increased number of biological replicates), which revealed significant changes in Atoh1 and Id4 expression in response to FST overexpression (Figure 3 E). We added a quantification of normalized p-Smad2/3 levels.

Interestingly, the authors show that Follistatin overexpression appears to delay cell cycle exit in an apical-to-basal direction. And also, claim to show a normal medial-to-lateral gradient of cell cycle exit that is also opposed by follistatin overexpression. However, the data presented in Figure 4I is very difficult to understand, perhaps because of the way it is graphed. It might be useful for the authors to show these data from IHCs and OHCs on two separate graphs (with associated statistics). Nonetheless, the numbers seem to indicate that follistatin increases the percentage of OHCs that take up EdU in the mid-cochlea relative to IHC, suggesting differential sensitivity from medial-to-lateral.

As requested, in the revised manuscript OHC and IHC EdU incorporation is now graphed separately. We are puzzled by the reviewer’s conclusion that follistatin overexpression increases the percentage of OHCs that take up EdU in the midcochlea compared to IHCs, as this is not what our data indicates.

However, this result appears to contrast with the result in the earlier part of Figure 4, which shows the ectopic proliferation in the IHC region to be more robust? Whereas, in Figure 5, where there appears to be an increase in ectopic inner HCs, accompanied by either normal or depleted three rows of outer hair cells in FST-OE. I would have expected, based on Figure 4K, that the opposite might be true

See comment above.

Finally, in Figure 5, the authors use an Inhba knockout strategy, and discover that there are sporadic ectopic inner HCs present in the KO, with an increase from base-to-apex. However, given the increase in proliferation seen in the follistatin OE, one might expect both IHC and OHC to be affected, and that the effect would be stronger and less organized than that shown in Figure 5Q-S. The phenotype shown, that of ectopic IHC occurring in a single row, seems to indicate a problem with border formation, rather than an overproliferation phenotype?

We agree with the reviewer that defects in border formation could attribute to ectopic IHCs. However, as shown in much detail we observe that FST overexpression disrupts the medial-to-lateral gradient of cell cycle exit, disrupting the proportion of inner and outer hair cell that are being produced.

Reviewer #3:[…] Essential revisions:In McCullar et al., 2010, the study demonstrated that the addition of Activin A to post-hatch avian auditory sensory epithelium increases proliferation while inhibiting the Activin receptors Acvr2a and 2b results in a decrease in proliferation in the sensory epithelium. This significantly lessens the novelty/impact/advance of the current study. It is unfortunate that the authors did not reference this paper.

McCullar et al. investigated the role of Activin signaling in the mature chicken auditory organ. We disagree with the reviewer’s view that the aforementioned study lessens the impact of our current study. The study was conducted in explanted adult chicken tissue examining the role of Activin signaling in regenerative supporting cell proliferation-a process that does not exist in mammals. More importantly, Activin signaling in mature chicken auditory sensory epithelium promotes supporting cell (progenitor cell) proliferation, whereas in the developing murine cochlea Activin signaling promotes progenitor cell differentiation. The striking differences in Activin’s function in the avian and mammalian auditory organ are quite intriguing and we added a brief discussion of the different roles of Activin signaling in birds and mammals in the Discussion section.

Furthermore, in the 2010 study McCullar et al., also demonstrated the expression patterns of the Activin receptors Acvr2a and 2b as well as the downstream Smad effector proteins pSmad2 and pSmad1/5/8 in the mammalian cochlea and vestibule (Figure 11, Figure 12, Figure 13 and Figure 14 and Table 4).

The mentioned expression data relates to adult stages and thus has little bearing on the role of Activin signaling during development.

The expression patterns presented in Figure 1 have been previously published as well. Specifically, the expression of Inhba and Fst have been published by Son et al., 2015 at E13.5 (Figure S5), E15.5 (Figure 5) and E18.5 (Figure S6) clearly demonstrating the spatiotemporal basal-to-apical expression of these two molecules. The Atoh1 and Sox2 expression patterns have also been previously published by many groups including the current authors. Thus, Figure 1 does not offer any significant new information.

The purpose of Figure1 is to characterize Atoh1, Inhba and Fst expression at the onset of hair cell differentiation, which the aforementioned study did not provide.

[Editors' note: the author responses to the re-review follow.]

The authors have done a very nice job of addressing the reviewers' comments. There are a few remaining suggestions from the reviewers. The further experimental work suggested by reviewer 1 is left to the authors' discretion.

We thank our two reviewers and our editors Marianne Bronner (Senior Editor) and Tanya Whitfield (Reviewing Editor) for the favorable evaluation of our revised article. Please see below our response to the reviewer’s suggestions:

Reviewer #1:This is a re-submission by Prajapati-DiNublia et al., examining the role of Activin A in hair cell differentiation.I was not suggesting electrophysiological experiments which I agree would be beyond the scope of this study; rather FM1-43 uptake which would provide evidence of channel/stereociliary function. Furthermore, performing immunohistochemistry to demonstrate innervation phenotypes (if any) is straight forward and would add value to the study.Lastly, I do agree with the other reviewer that quantification of P-SMAD should include controls showing total receptor protein using antibodies against the un-phosphorylated SMAD form. This would control for changes in total protein levels.

We appreciate the reviewer’s suggestions. As the suggested experiments are not essential and to not further delay the publication of the manuscript, we decided to not conduct additional experimental work.